# Dynamic Model Routing and Cascading for Efficient LLM Inference: A Survey

## Abstract

The rapid growth of large language models (LLMs) with diverse capabilities, costs, and domains has created a critical need for intelligent model selection at inference time. While smaller models suffice for routine queries, complex tasks demand more capable models. However, static model deployment does not account for the complexity and domain of incoming queries, leading to suboptimal performance and increased costs. Dynamic routing systems that adaptively select models based on query characteristics have emerged as a solution to this challenge.

This survey provides a systematic analysis of multi-LLM routing and cascading approaches, focusing on systems that route queries across a pool of independently trained LLMs at inference time. We cover diverse routing paradigms, including query difficulty, human preferences, clustering, uncertainty quantification, reinforcement learning, multimodality, and cascading. For each paradigm, we analyze representative methods and examine key trade-offs. Beyond taxonomy, we introduce a conceptual framework that characterizes routing systems along three dimensions: when decisions are made, what information is used, and how they are computed. This perspective highlights that practical systems are often compositional, integrating multiple paradigms under operational constraints.

Our analysis demonstrates that effective multi-LLM routing requires balancing competing objectives. Choosing the optimal routing strategy depends on deployment and computational constraints. Well-designed routing systems can outperform even the most powerful individual models by strategically leveraging specialized capabilities across models while maximizing efficiency gains. Meanwhile, open challenges remain in developing routing mechanisms that generalize across diverse architectures, modalities, and applications.

# 1 Introduction

## 1.1 Problem and Motivation

Production deployment of large language models faces a fundamental cost-performance dilemma. Queries vary substantially in complexity, from simple factual questions to complex multi-step reasoning problems. When a single model handles all requests, simple queries consume unnecessary resources when routed to powerful models, while complex queries may exceed the capabilities of smaller models.

Recent advances in multi-LLM deployment have introduced dynamic routing systems that address this challenge. These systems analyze each query and select from a pool of models with different capabilities, costs, and specializations. By matching computational resources to query requirements, adaptive routing can reduce costs while maintaining or improving output quality.

## 1.2 Routing and Cascading

This survey covers two complementary approaches to adaptive model selection at inference time:

**Model routing** analyzes each input and selects the most appropriate model based on query characteristics. The router makes a single decision, mapping the query to one model from the available pool.

**Model cascading** operates sequentially, first attempting inference with smaller, faster models and escalating to larger, more capable models only when the initial response is deemed insufficient based on quality estimation.

Both approaches aim to optimize the performance-cost trade-off by matching queries to appropriate models. Production systems often combine routing and cascading strategies to maximize efficiency.

## 1.3 Scope and Organization

This survey focuses on routing between independently trained LLMs at inference time, where the goal is to select the most suitable model from a discrete pool given a query. In all cases, the routing mechanism is treated as a separate orchestration layer, distinct from the candidate models it selects among. We organize methods into six main paradigms based on their routing strategy, while acknowledging that methods may span multiple categories:

- **Difficulty-aware routing** (Section 2): Routes based on estimated query complexity

- **Human preference-aligned routing** (Section 3): Leverages preference data from human feedback

- **Clustering-based routing** (Section 4): Groups similar queries using unsupervised learning

- **Reinforcement learning routing** (Section 5): Learns routing decisions through reinforcement learning, including policy optimization (e.g. PPO, GRPO) and bandit methods

- **Uncertainty-based routing** (Section 6): Routes based on model confidence estimates

- **Cascading** (Section 7): Sequential multi-model approaches

Additionally, we briefly explore research on routing of multimodal models (Section 8), and cover evaluation approaches, including benchmarks and metrics (Section 9). We then synthesize the surveyed methods through a multidimensional framework in Section 10. Finally, we discuss open challenges and future directions (Section 11), providing a foundation for advancing research in efficient multi-LLM deployment.

### 1.4 Conceptual Design Space for LLM Routing

The paradigms we cover in this survey (cf. Section 1.3) provide a useful foundation for organizing and understanding the literature. In practice, real-world systems draw on more than one of these paradigms simultaneously. To complement the paradigm-based organization, routing approaches can be categorized through a multidimensional conceptual framework.

**When the routing decision is made.** Routing systems can rely on either pre-generation or post-generation decisions, or they can adopt a multi-stage process. Pre-generation routing selects a model before generating any output, relying entirely on properties of the incoming query, while post-generation routing systems make their decision after an initial response has been produced, using output quality or confidence as the primary signal. Instead of making a one-time decision, some approaches embed routing within a sequential multi-stage process, escalating across models based on response quality. Multi-stage systems are not mutually exclusive with pre- or post-generation routing; they typically combine both within a sequential escalation structure.

**What information the routing mechanism uses.** Routing systems differ substantially in the richness of their input signals. The simplest approaches operate on the query alone, using lexical or semantic features to characterize the request. More informed systems additionally incorporate metadata about available models to guide selection, such as cost, latency, or domain specialization. Post-generation approaches further extend this by incorporating response-level signals, such as confidence scores, token probabilities, or verifier outputs. Some systems also accumulate external feedback over time, adapting their routing behaviour based on user interactions or downstream task performance.

**How the decision is computed.** Routing decisions vary considerably in their computational complexity. At one end, simple threshold rules or cost-based heuristics require no training and can be applied directly at inference time. At the other end, supervised classifiers trained on historical performance data learn to predict which model is likely to handle a given query best. More sophisticated approaches either employ bandit algorithms that update routing behaviour through ongoing interaction at deployment time, or reinforcement learning methods such as PPO and GRPO that optimize a routing policy during training. In practice, many systems combine these mechanisms, for example using a classifier to make an initial routing decision and a threshold rule to trigger escalation within a cascade.

These dimensions are not independent of the paradigms. Table 1 illustrates this by mapping representative methods from each of the six paradigms to these dimensions. As Table 1 illustrates, clustering and difficulty-aware methods are generally pre-generation approaches that operate on query-level signals, while uncertainty-based methods and cascades typically involve post-generation decisions on response-level signals. Interestingly, cascading approaches tend to employ layers of these individual techniques for diverse objectives, ranging from balancing performance-cost trade-offs to implementing safety measures on input queries and output generations. This multidimensional view helps reveal where systems genuinely overlap, as real-world deployments rarely conform to a single paradigm. Production systems typically combine multiple routing strategies, adapting their behaviour to the demands of diverse queries, operational constraints, and evolving user needs. This framework therefore offers a unifying lens that complements the paradigm-based organization of the sections that follow. Section 10 revisits this framework after the paradigm-based survey, examining how production systems compose across these dimensions and identifying structural gaps in the literature.

## 2 Difficulty-aware Routing

When an LLM is able to answer a query correctly, it is often because the query is relatively simple or closely aligned with the model's training data. On the contrary, more complex queries such as maths or sophisticated coding tasks may require advanced reasoning capabilities. Driven by this observation that some queries are inherently more "difficult" than others, researchers have explored the idea of routing queries to different LLMs based on their difficulty level.

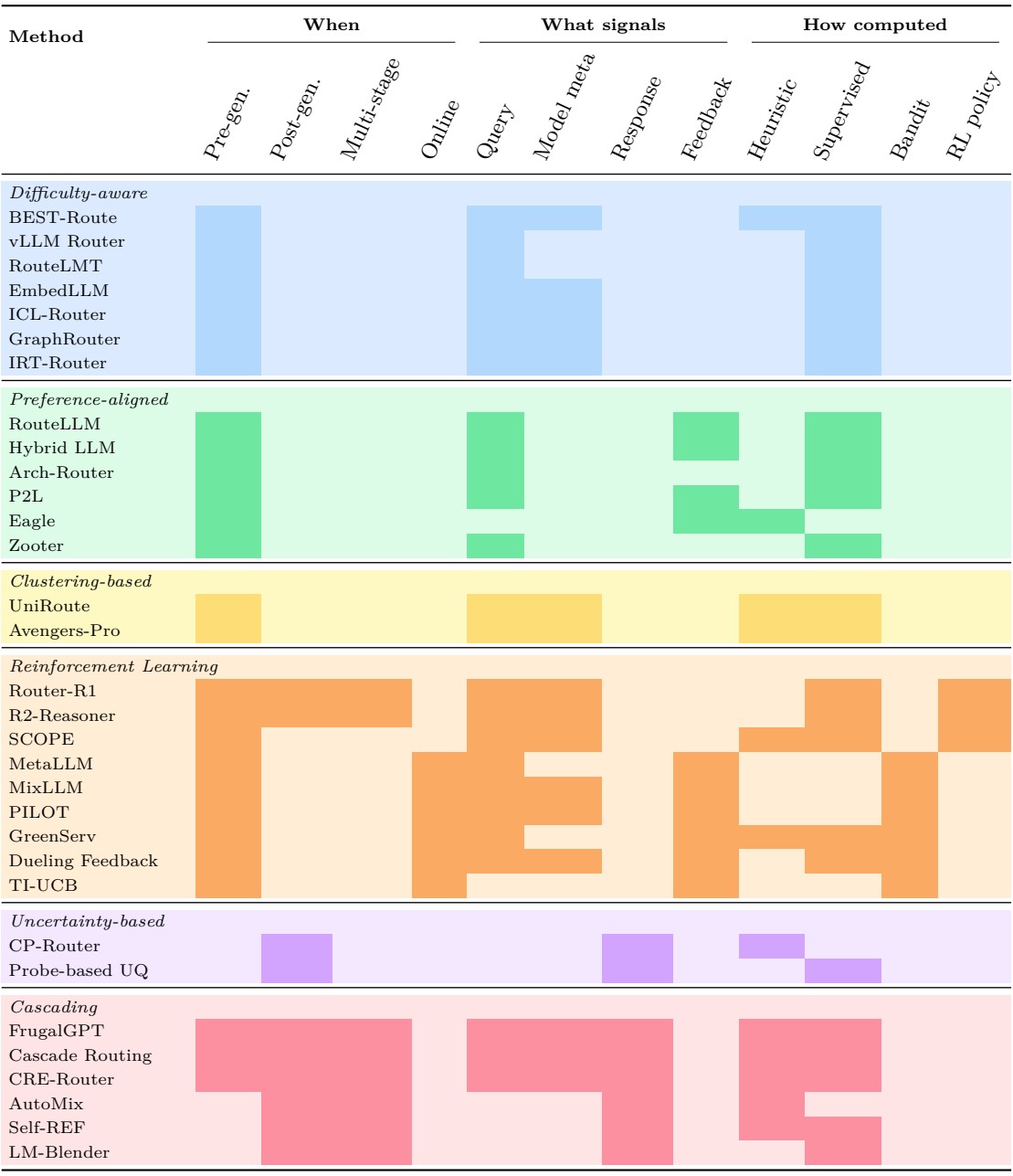

Table 1: Design-space matrix mapping representative routing and cascading methods to the survey's conceptual framework along three axes: *when* routing decisions are made (Pre-gen., Post-gen., Multi-stage, Online/adaptive), *what* signals are used (Query, Model metadata, Response, Feedback), and *how* decisions are computed (Heuristic, Supervised, Bandit, RL policy). Shaded cells indicate the property applies; cell and row colour identify the paradigm group. Multi-stage denotes a sequential escalation structure and is not mutually exclusive with Pre-gen. and Post-gen., which describe the types of decisions made within that structure. Multiple filled cells in the *how* group indicate that a method combines mechanisms, e.g. a learned classifier paired with a threshold rule.

Query difficulty can be assessed using various methods, including heuristic-based approaches (e.g. text length, word rarity, idiomatic language, syntactic complexity), learned classifiers, or even leveraging the LLMs themselves to estimate the complexity of a query, usually referred to as "LLM as a judge" (Proietti et al., 2025). Once the difficulty of a query is estimated, it can be routed to an appropriate LLM. For instance, simpler queries can be directed to smaller, more cost-effective models, while complex queries can be sent to larger, more capable models. This optimizes resource allocation while maintaining performance. In this sense, several routing methods can fall under this dynamic model selection paradigm, tailored to query difficulty. Implementations can prioritize different objectives, such as cost, latency, accuracy, or human preference. Within the design-space framework (cf. Table 1), difficulty-aware methods are characteristically pre-generation approaches that operate on query-level signals, typically implemented as supervised classifiers or heuristics.

Among the work that can be classified under the difficulty-aware routing paradigm is BEST-Route (Ding et al., 2025) which dynamically allocates queries to different LLMs depending on their assessed difficulty. BEST-Route uses a multi-head router based on DeBERTa-v3-small (He et al., 2021; 2023) to estimate query difficulty and select the optimal model and sampling strategy. At inference time, it employs best-of-$n$ sampling (Stiennon et al., 2020; Nakano et al., 2021) to enhance the performance of small models, selecting the best response out of $n$ generated responses, using a proxy reward model fine-tuned from OpenAssistant RM, which is based on a DeBERTa-v3-large. If necessary, sampling is increased, or harder queries are routed to a larger model. At inference time, users can set thresholds to balance cost and accuracy, where higher thresholds favour the reference model, improving quality at increased costs. Since multiple small-model and best-of-$n$ combinations can meet a given threshold, BEST-Route effectively selects among them using cost estimation to ensure high-quality responses at minimal cost. In the area of machine translation, Wu et al. (2025b) apply a similar intuition, proposing source-side routing policies that decide whether to use an NMT system or an LLM based on the complexity of the source sentence.

Similarly, vLLM Semantic Router (Wang et al., 2025a) addresses the challenge of adaptive reasoning, where costly step-by-step inference is applied only when beneficial, while maintaining low latency and efficiency for straightforward queries. The system uses a ModernBERT-based classifier (Warner et al., 2025) to analyze query intent and complexity, routing queries that require reasoning to models with chain-of-thought capabilities while directing simpler queries to standard inference without reasoning. This reasoning-aware approach demonstrates that selective application of computational resources based on query complexity can simultaneously improve both accuracy and efficiency.

Recently, RouteLMT (Luo et al., 2026) has proposed an in-model router that probes the small translator's prompt-token representations via lightweight LoRA adaptation, without requiring external models or hypothesis decoding. It identifies the marginal gain, i.e., the large model's improvement over the small model, as the optimal signal for routing decisions.

In a relevant line of work, EmbedLLM (Zhuang et al., 2024) employs matrix factorization to learn compact vector embeddings of LLMs, capturing model characteristics such as domain specialization. At routing time, these embeddings are used to predict which model will answer a given query correctly, effectively estimating query-model compatibility without requiring explicit difficulty labels. The approach enables efficient routing and benchmark accuracy prediction, though it requires retraining when incorporating new models.

ICL-Router (Wang et al., 2026) represents model capabilities as compact in-context vectors, derived from a small set of query-performance pairs that reflect each model's capability profile. An LLM-based router is trained in two stages, first to align query embeddings with its semantic space, then to predict whether each model can correctly answer a given query using these vectors as in-context input. A key advantage is that new models can be integrated without retraining the router, by simply evaluating them on a representative query set to generate their capability profile.

GraphRouter (Feng et al., 2024) takes a different approach by modelling the relationships among tasks, queries, and LLMs through a heterogeneous graph structure. The method constructs a graph with task, query, and LLM nodes, where edges represent interactions between these entities. A Graph Neural Network (GNN) trained on historical performance and cost data performs edge prediction to forecast both the effectiveness and expense of routing a query to a specific LLM. This inductive learning framework enables

GraphRouter to generalize to newly introduced LLMs without retraining, as it learns patterns from node characteristics rather than memorizing specific models. Experiments demonstrate substantial performance improvements while reducing computational overhead.

IRT-Router (Song et al., 2025) explicitly models two complementary properties for routing: the ability of each LLM, learned from its profile metadata, and the difficulty and discrimination of each query, extracted from its embedding. These are combined in an interactive layer to predict the probability that each candidate LLM will correctly answer the query. The predicted performance is then combined with each LLM's cost to produce a routing score, and the query is directed to the highest-scoring model. This formulation, inspired by Item Response Theory from educational psychometrics, provides interpretable routing decisions and handles unseen queries at deployment time through a semantic similarity warm-up mechanism.

While our survey focuses on multi-LLM routing, it is worth mentioning that difficulty estimation has also been explored in single-LLM contexts. Recent research found that LLMs frequently "overthink" simple queries and "underthink" complex ones (Aggarwal et al., 2026), leading to inefficiencies. Moreover, sophisticated chain-of-thought reasoning primarily improves performance on complex maths and logic tasks, with limited gains for other tasks (Wei et al., 2022; Sprague et al., 2025). On one hand, difficulty estimation can be calculated before sending the query to a reasoning LLM to decide whether to activate the thinking mode[1] (Zhang et al., 2025b;d), or to constrain the thinking token budget (Shen et al., 2025). On the other hand, probing the LLM during the reasoning process can help identify whether to continue with further reasoning steps or to stop early (Wu et al., 2025a; Jiang et al., 2025). This line of work can fall under adaptive computation approaches that aim to optimize resource usage within a single LLM.

## 3 Human Preference-aligned Routing

In this paradigm, human preference data is used to train routers that generalize across domains and LLMs. Within the design-space framework (cf. Table 1), these methods are characteristically pre-generation approaches that operate on query-level preference signals, with routing decisions computed via supervised classifiers or win-prediction models. For example, RouteLLM (Ong et al., 2025) introduces a learning framework for training router models that dynamically select between strong and weak LLMs at inference time, optimizing the trade-off between response quality and cost. It formulates routing as a binary decision between strong (high-quality, high-cost) and weak (lower-quality, low-cost) LLMs, and employs a win prediction model to estimate the probability that the strong model will outperform the weak model for a given query. RouteLLM optimizes router parameters by maximizing the likelihood of observed preference data. Unlike Hybrid-LLM (Ding et al., 2024) and Arch-Router (Tran et al., 2025) that use solely synthetic preference data generated by an LLM, RouteLLM (Ong et al., 2025) leverages human preference labels from Chatbot Arena (Chiang et al., 2024). In addition, RouteLLM authors obtain pairwise comparison labels using an LLM judge. This data augmentation technique with synthetic preference labels substantially improves router performance. They experiment with four router architectures, namely similarity-weighted SW Ranking, matrix factorization, BERT, and causal LLM (based on Llama 3 8B). The matrix factorization router and causal LLM routers perform very competitively on MT Bench (Zheng et al., 2023) when trained on the augmented dataset of both Chatbot Arena and LLM judge, with matrix factorization being the most efficient and cost-effective router.

Similarly, Arch-Router (Tran et al., 2025) aligns routing decisions with explicit user preferences. However, rather than optimizing purely for cost or performance, it allows users to define domain-action pairs. For instance, legal summarization queries can be routed to one model, while code generation requests will be sent to another model. The key insight is that routing policies are provided as part of the input context to a 1.5B parameter model, which means users can update their preferences without retraining. This is particularly valuable in production environments where routing requirements evolve frequently. Nevertheless, this flexibility comes with trade-offs, as a 1.5B model introduces additional computational overhead compared to

---

[1] Some approaches such as AdaptThink (Zhang et al., 2025b) decide whether to activate the thinking mode from the beginning, while other approaches such as Continue-Thinking (Zhang et al., 2025d) first estimate the quality of the generated response without thinking to decide whether to continue generation with thinking.

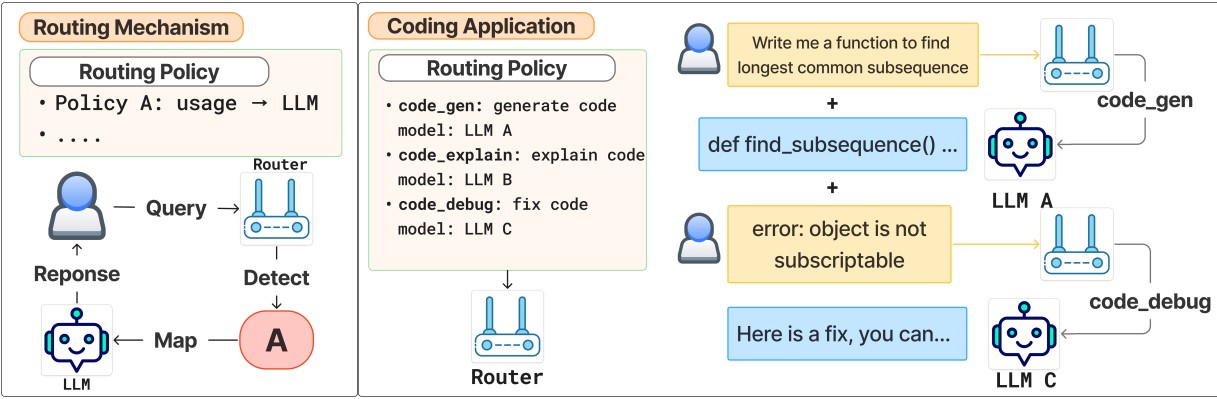

Figure 1: Arch-Router's Preference-Aligned Routing Mechanism. The routing policies and user conversation are provided to the router to select the appropriate policy and corresponding LLM. Example usage in coding is shown on the right.

lightweight classification-based routers, which may be suboptimal in latency-sensitive applications. Figure 1 illustrates the preference-aligned routing mechanism that Arch-Router has introduced.

Prompt-to-Leaderboard (P2L) (Frick et al., 2025) extends traditional preference-based leaderboard evaluation by training an LLM to generate prompt-specific Bradley-Terry coefficients for model ranking. Unlike standard leaderboards that average performance across all tasks, P2L takes a user's prompt as input and outputs customized strength scores for each model, predicting which LLM will perform best for that particular query. The method enables task-specific evaluation, optimal cost-constrained routing, and personalized model selection based on user history.

Eagle (Zhao et al., 2024) introduces a training-free approach to preference-aligned routing by using ELO ranking systems (Elo, 1967). The method combines two complementary modules: Eagle-Global, which evaluates model overall capability using ELO ratings across the entire historical dataset, and Eagle-Local, which assesses specialized abilities through ELO rankings on similar past queries. By transforming sparse pairwise comparisons into comprehensive rankings, Eagle efficiently integrates user feedback without requiring model training.

While the aforementioned methods directly optimize against human or synthetic preference comparisons, some approaches rely on reward models trained on such data to provide supervisory signals. Zooter (Lu et al., 2024) takes a reward-guided supervised learning approach, training a routing classifier using labels derived from the QwenRM reward model (Bai et al., 2023), which scores candidate LLM responses on training queries.[2] The routing function outputs a categorical distribution over a pool of LLMs, representing the likelihood that each model can best handle the query. This knowledge distillation approach employs a BERT-style router (mDeBERTa-v3-base (He et al., 2023)). However, since training relies on a fixed set of LLMs, Zooter's adaptability to new or unseen models is limited.

RouterDC (Chen et al., 2024b) addresses the same task via dual contrastive learning, training a shared encoder alongside per-LLM embedding vectors. A sample-LLM contrastive loss pulls each query toward the embeddings of LLMs that answer it correctly while pushing away those that do not; a complementary sample-sample contrastive loss groups semantically similar queries together, with groups defined by K-means clustering on the training set. At inference, routing reduces to selecting the LLM whose embedding is most similar to the query representation, a single forward pass with no LLM calls until the final selected model.

---

[2]QwenRM is typically used for RL training (e.g., GRPO) or SFT data construction through rejection sampling. Zooter uses it solely to generate routing labels.

# 4 Clustering-based Routing

Clustering-based routing groups similar queries using unsupervised learning and assigns each cluster to the most suitable LLM. This approach balances performance and cost by routing different query types to their most cost-effective models, without requiring explicit task labels.

As illustrated in Figure 2, UniRoute (Jitkrittum et al., 2026) applies $K$-means clustering to an unlabelled training dataset to identify $K$ centroids, then partitions a validation set into these representative clusters. Each candidate LLM is evaluated on the validation data within each cluster, with scores adjusted by a predefined cost value for that model. At inference time, incoming queries are compared against the cluster centroids to determine which LLM should handle the request, with the accuracy-cost trade-off controlled through dynamic cost adjustment during routing.

Among the main practical advantages of this clustering approach is the ability to add new LLMs at inference time without retraining, as they are simply evaluated on existing clusters to obtain their performance profile. Moreover, the method can operate entirely without task labels, relying solely on query embeddings and cluster assignments. UniRoute demonstrates effective scaling, handling routing decisions across unseen LLMs.

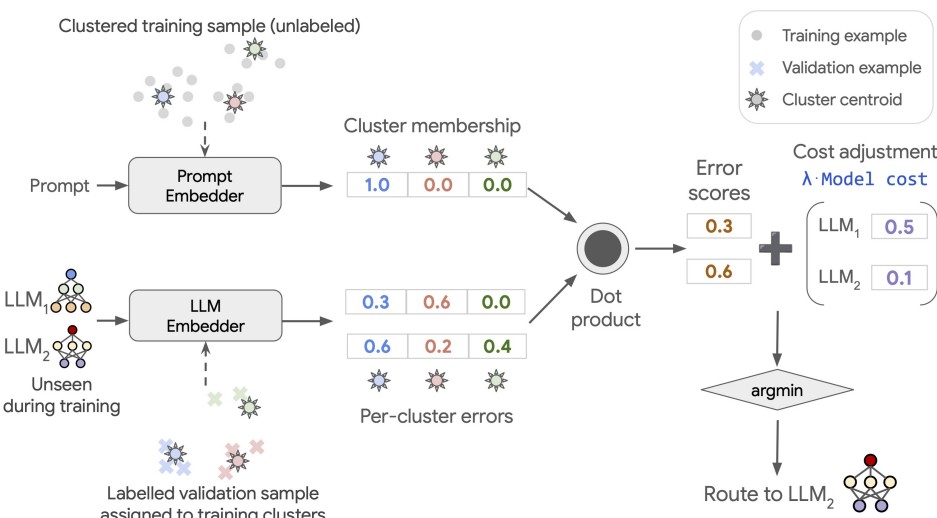

Figure 2: UniRoute cluster-based router. First, perform $K$-means on a training set to find K centroids, and then partition the validation set into K representative clusters. Each test-time LLM can then be represented as a $K$-dimensional feature vector of per-cluster errors. This yields an intuitive routing rule: for each test prompt, route to the LLM with the smallest cost-adjusted average error on the cluster the prompt belongs to. The prompt embedder may either be completely unsupervised (as shown in the figure), or fitted via supervised learning using labels from a set of training LLMs different from those seen during test time.

The Avengers (Zhang et al., 2026) introduces a general clustering-based routing recipe for combining smaller open-source models into a collective ensemble. Incoming queries are embedded and grouped into semantic clusters; each candidate model is scored per cluster during training; and at inference time each query is assigned to its nearest cluster and routed to the top-performing model(s), with repeated sampling and majority voting to aggregate outputs. This four-step paradigm—embed, cluster, rank, route—demonstrates that coordinated ensembles of smaller models can match or exceed the accuracy of proprietary models without modifying any individual model. Avengers-Pro (Zhang et al., 2025e) extends this paradigm by incorporating an explicit performance-efficiency trade-off. Each cluster assignment is governed by a scoring function that interpolates between correctness and inference cost via a single hyperparameter, tracing an accuracy-cost Pareto frontier across operating points.

Within the design-space framework (cf. Table 1), clustering-based methods are characteristically pre-generation approaches operating on query-level signals, with routing decisions made by unsupervised assignment to learned centroids. CRE-Router (Moslem et al., 2026) demonstrates that this paradigm can be integrated into multi-stage systems, proposing a two-stage cascaded solution where Stage 1 clusters incoming queries and assigns each cluster to its most cost-effective model, and Stage 2 adds a quality estimation cascade to escalate low-quality outputs to a stronger model.

## 5 Reinforcement Learning Routing

Reinforcement learning (RL) approaches formulate routing as decision-making under uncertainty. Methods in this section include policy optimization algorithms that iteratively refine routing decisions through multi-step interactions and online bandit algorithms that balance exploration and exploitation with real-time feedback. Within the design-space framework (cf. Table 1), policy optimization methods make multi-stage decisions drawing on query and model-level signals, while bandit methods are pre-generation approaches that adapt online through deployment feedback; both occupy the more complex end of the *how* axis.

### 5.1 Policy Optimization Methods

Policy optimization approaches such as Router-R1 (Zhang et al., 2025a) and Route-and-Reason (R2-Reasoner) (Shao et al., 2025) optimize routing decisions through multi-step interactions. These RL approaches leverage the strengths of multiple LLMs to solve complex reasoning tasks while optimizing performance-cost trade-offs. While Router-R1 addresses multi-round routing and aggregation, R2-Reasoner focuses on subtask-level decomposition. Although the router in these methods is itself an LLM performing multi-step reasoning, all methods in this section dispatch queries to independently trained, separately deployed models, so the routing happens between model boundaries rather than within a shared architecture.

Router-R1 formulates routing as a sequential decision-making process. It alternates between performing internal reasoning ("think" action) and assigning a specific model from a pool of available LLMs ("route" action), iteratively refining answers. The router itself is an LLM that can leverage its reasoning ability to dynamically select and aggregate responses from multiple models. Router-R1 initiates a maximum of 4 routing steps per input query. The authors use Qwen2.5-3B-Instruct (Yang et al., 2024) and LLaMA-3.2-3B-Instruct (Dubey et al., 2024) as base models. They are trained with the Proximal Policy Optimization (PPO) (Schulman et al., 2017) reinforcement learning algorithm, on a training dataset of question–score pairs representing the quality of responses of each LLM. The authors argue that Router-R1 can generalize to new LLMs without retraining by conditioning on simple descriptors such as pricing and latency.

Similarly, R2-Reasoner decomposes complex reasoning tasks into simpler subtasks using a dedicated Task Decomposer. These subtasks are then allocated to the most suitable model by a Subtask Allocator. R2-Reasoner employs a staged training pipeline, combining supervised fine-tuning and reinforcement learning with Group Relative Policy Optimization (GRPO) (Shao et al., 2024). This iterative approach refines the decision-making capabilities of the decomposer and allocator without requiring end-to-end gradient propagation. The authors argue that the framework achieves substantial cost reductions while maintaining competitive reasoning accuracy. It also generalizes well to unseen models, making it adaptable to dynamic real-world scenarios.

Although both Router-R1 and R2-Reasoner aim to optimize cost, reasoning steps can incur multiple model calls, introducing inference latency. In this sense, these approaches are best suited for scenarios where cost savings outweigh latency concerns, such as routing to commercial LLM APIs with high per-call costs.

SCOPE (Scalable and Controllable Outcome Performance Estimator) (Cao et al., 2026) takes a distinct approach by using GRPO to train a performance estimator rather than a routing policy. Given a query and a candidate model, SCOPE retrieves behavioural examples from a pre-computed model fingerprint and predicts both the model's correctness and token cost via chain-of-thought reasoning. The final routing decision is then made by a simple rule-based selection using a user-defined accuracy–cost trade-off, requiring no further learning. Unlike Router-R1 and R2-Reasoner, which use RL to directly optimize routing decisions, SCOPE's RL component is closer to reward-shaped prediction training than policy optimization. Furthermore, by

characterizing models through behavioural fingerprints rather than fixed identifiers, SCOPE generalizes to unseen models without retraining.

## 5.2 Bandit-Based Methods

Bandit-based methods frame routing as an online learning problem, balancing exploration of model capabilities with exploitation of known strengths while adapting to feedback over time.

MetaLLM (Nguyen et al., 2025) formulates routing as a multi-armed bandit problem, dynamically selecting the least expensive LLM likely to provide a correct answer for each query. MetaLLM does not rely on reward models for training. Instead, it optimizes routing decisions based on an accuracy-cost trade-off, adapting to query difficulty and model performance over time.

MixLLM (Wang et al., 2025b) employs a contextual bandit framework with policy gradient methods to optimize query-LLM assignments. The system enhances query embeddings with domain-aware tags through unsupervised fine-tuning, enabling more accurate predictions of LLM-specific response quality and cost. A meta decision-maker selects assignments to balance quality, cost, and latency constraints. During online deployment, MixLLM updates its routing policy using binary user feedback through a dynamic feedback score mechanism, applying policy gradient optimization to maximize expected reward. This continual learning approach allows the system to adapt to evolving queries and changing LLM candidate sets.

PILOT (Preference-prior Informed LinUCB for Adaptive Routing) (Panda et al., 2025) extends contextual bandits by integrating offline human preference data with online evaluative feedback. The method reformulates routing as a contextual bandit problem where the system selects LLMs based on binary success/failure signals rather than full supervision. Building on the LinUCB algorithm (Li et al., 2010) widely used in recommender systems, PILOT incorporates preference priors from sources like Chatbot Arena to accelerate learning. The authors additionally model cost constraints as an online multi-choice knapsack problem (Chakrabarty et al., 2008) to ensure resource-efficient routing under budget limits. This combination enables dynamic, personalized LLM selection that balances performance and cost.

GreenServ (Ziller et al., 2026) extends the contextual bandit framework with an explicit focus on energy efficiency. Rather than optimizing for cost proxies such as API prices or token budgets, GreenServ measures GPU energy consumption directly during inference. It extracts a lightweight context vector per query, encoding task type, semantic cluster, and textual complexity, and then employs LinUCB (Li et al., 2010) to learn routing policies online across a pool of 16 open-access LLMs. A tunable trade-off parameter $\lambda$ allows operators to interpolate between accuracy-first and energy-first policies.

LLM Routing with Duelling Feedback (Chiang et al., 2025) addresses routing through a contextual duelling bandit formulation, learning from pairwise preference comparisons rather than absolute performance scores (cf. Section 3). The method introduces Category-Calibrated Fine-Tuning (CCFT), which constructs model embeddings by first fine-tuning text encoders with contrastive learning to cluster queries by category, then computing weighted combinations of category embeddings based on each LLM's domain expertise and cost profile. These learned embeddings enable the deployment of Feel-Good Thompson Sampling for Contextual Duelling Bandits (FGTS.CDB), a theoretically grounded Bayesian algorithm for online model selection. Evaluations on RouterBench and MixInstruct demonstrate that the approach achieves lower cumulative regret and faster convergence than baselines using general-purpose embeddings, while maintaining robust performance-cost balance.

Similarly, Time-Increasing UCB (TI-UCB) (Xia et al., 2024) addresses online model selection through a time-increasing bandit algorithm that accounts for the increasing-then-converging performance trend observed during iterative model fine-tuning, a common pattern as LLMs are progressively improved. Unlike traditional bandit methods that assume stationary reward distributions, TI-UCB incorporates a change detection mechanism to identify convergence points, enabling more efficient exploration-exploitation balance. The approach achieves logarithmic regret bounds and demonstrates practical effectiveness in both classification model selection and online LLM selection scenarios.

# 6 Uncertainty-based Routing

Researchers have been investigating whether there is a reliable way to estimate if a model is confident about its own response, where such confidence must be aligned with actual correctness. This notion has several applications in routing, as effective uncertainty quantification enables systems to identify when to escalate queries to more capable models. Uncertainty-based methods are post-generation approaches that use response-level signals, i.e. confidence scores or token probabilities, to trigger escalation decisions.

## 6.1 Probing and Probability-based Uncertainty Estimation

In their work, Chuang et al. (2025b) benchmark eight uncertainty quantification methods for routing from SLMs to LLMs on edge devices. They observe that probe-based methods (trained classifiers) and perplexity-based methods outperform verbalization approaches (self-reported confidence). They demonstrate that SLMs match LLM performance on high-confidence (top 20%) queries. These results are in line with the experiments by Mahaut et al. (2024) who demonstrate that trained hidden-state probes provide more reliable confidence estimates than alternatives, even though at the expense of requiring access to weights and supervision data.

CP-Router (Su et al., 2025) applies Conformal Prediction to route between standard LLMs and Large Reasoning Models (LRMs) like DeepSeek-R1, which often generate verbose reasoning for simple queries. For multi-choice question answering, the method extracts logits for answer options, applies softmax to obtain a probability distribution, and computes uncertainty scores as 1 minus the probability for each option. It then builds a prediction set based on these scores against a calibration-derived threshold. Queries with a single plausible option (high confidence) are handled by the standard LLM, while those with multiple plausible options (high uncertainty) are routed to the more capable but costly LRM.

## 6.2 LLM-as-a-Judge Uncertainty Estimation

Instead of confidence calculations, some researchers explored using LLMs themselves to estimate the quality of either their own responses or those generated by other models.

Zhang et al. (2025c) propose a confidence-driven LLM router that leverages uncertainty estimation derived from response quality evaluation. Rather than estimating confidence from model-internal signals, the method employs an external LLM-as-a-Judge to simulate human rating preferences, jointly optimizing response quality and system cost for edge–cloud routing.

In contrast, other researchers employ self-verification or self-reporting, where an LLM is asked to report its own confidence. Chuang et al. (2025b) refer to these approaches as "verbalization" and classify them as one-step verbalization, where an LLM is prompted to output both the answer and numeric confidence in one step, and two-step verbalization, where the confidence score is obtained in a separate, follow-up query after the model has provided an answer (Tian et al., 2023). However, their experiments indicate that both verbalization techniques consistently exhibit low alignment between reported uncertainty and prediction correctness.

Despite these limitations, recent work demonstrates that self-verification can be effective in cascading systems when combined with other mechanisms. AutoMix (Aggarwal et al., 2024) uses few-shot in-context learning for self-verification, while Self-REF (Chuang et al., 2025a) employs lightweight fine-tuning to improve confidence calibration. These methods are discussed in detail in Section 7 on cascading approaches, as they rely on self-verification as part of multi-stage routing pipelines rather than as standalone uncertainty quantification.

# 7 Cascades

While routing preselects an appropriate LLM for a query, cascading sequentially queries a pool of LLMs until a reliable response is obtained. Cascading strengthens the routing process by allowing multiple LLMs to be involved in answering a single query. In a typical cascading setup, a smaller and cheaper LLM is first queried to generate an initial response. Based on the quality of this response, the system decides whether to accept it or escalate the query to a larger and more capable LLM for further refinement or regeneration. This

cascading system design can be particularly useful when dealing with queries of varying difficulty levels, as it allows the system to adaptively allocate resources based on the complexity of the task at hand. Within the design-space framework (cf. Table 1), cascading systems are inherently multi-stage, combining query-level and response-level signals across successive models, and often compose multiple mechanisms within a single pipeline.

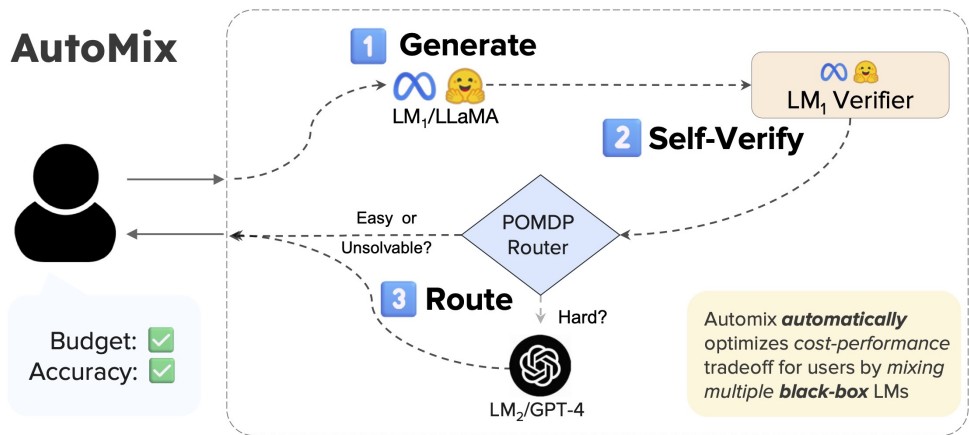

Figure 3: AutoMix example for two-model setup. Instead of relying only on a small model (SLM) with low performance or a large model (LLM) with high cost, AutoMix automatically mixes multiple black-box language models, based on user desired cost-quality tradeoff. AutoMix works in a 3-step process: (i) generation by a small model (LM1), (ii) self-verification of the generated answer, (iii) using confidence assessments from self-verification to do appropriate routing to a larger model (LM2). For N-model setup, the process is repeated till the final answer is reported.

FrugalGPT (Chen et al., 2024a) mixes routing and cascading through employing three main components: an LLM router, a threshold-based quality estimator, and a stop judge. Given a user query $q$, the router first selects an LLM to obtain its response to the query. However, in their setup, they adopted query-agnostic permutation for simplicity. Next, the quality estimator based on DistilBERT (Sanh et al., 2019) takes the query, the response, and the selected LLM as input and generates a quality score as output. Based on the quality estimation and the invoked LLM service, a cost-aware stop judge determines whether (i) to stop and return the answer, or (ii) to repeat the process of invoking the LLM router and generation scorer. Similarly, Dekoninck et al. (2025) propose cascade routing, a unified framework that integrates routing and cascading into a single system. Unlike pure routing, which selects a single model per query, or cascading, which runs models sequentially from smallest to largest, cascade routing iteratively selects the best model at each step, allowing it to skip or reorder models dynamically. The authors identify quality estimation as the critical factor for model selection success and show that cascade routing outperforms individual approaches.

CRE-Router (Moslem et al., 2026) illustrates how clustering and cascading can be composed into a unified pipeline in two pre-generation and post-generation stages. Stage 1 assigns incoming queries to semantic clusters and routes each cluster to its most cost-effective model using a scoring function that trades off error rate against inference latency, with the cost budget controlled by an interpretable hyperparameter tuned offline on task-correctness labels (cf. Section 4). Stage 2 then applies a quality estimation cascade, where a lightweight classifier inspects each efficient-model output and escalates low-quality responses to a stronger model. Collective pre-generation routing at the cluster level avoids running efficient models on query groups they are unlikely to handle correctly, while the post-generation quality estimation cascade recovers accuracy on individual queries where the efficient model falls short. Together, the two stages span the design space from pre-generation cluster-level routing to post-generation response evaluation, with the framework adapting to model pool changes through automatic Pareto analysis.

Instead of relying on an external model for quality estimation, some researchers have proposed "self-verification" as an efficient approach for model cascading (Aggarwal et al., 2024; Chuang et al., 2025a). In this paradigm, an efficient LLM learns to estimate the quality of its own response. The self-verification

score is then used to decide how to proceed. For instance, AutoMix (Aggarwal et al., 2024) uses a smaller model to generate an initial answer and self-verify its response before potentially routing the query to a larger model. The approach relies on few-shot prompting without fine-tuning, making it suitable for black-box models. While previous work finds self-verification unreliable to repair the model's output (Huang et al., 2024) or estimate its confidence (Mahaut et al., 2024; Chuang et al., 2025b), AutoMix demonstrates that few-shot self-verification provides a valuable signal that can be used for routing to an appropriate model. The final step involves using a router to decide whether to escalate the query to a larger model if it is a hard query, or output the current answer to the user if it is easy or unsolvable.[3] The router is modelled as a Partially Observable Markov Decision Process (POMDP) (Åström, 1965; Wu et al., 2021), which relies only on self-verification confidence as input. Figure 3 illustrates the AutoMix approach.

While AutoMix relies solely on few-shot in-context learning without any fine-tuning, Self-REF (Chuang et al., 2025a) introduces a lightweight fine-tuning strategy that enables LLMs to express confidence in their predictions using special confidence tokens. This confidence signal is then used for downstream routing, where queries with low confidence are routed to stronger (but more costly) LLMs, or rejection where the system should fall back on a safe default behaviour. Self-REF augments training data with confidence tokens based on correctness, fine-tunes the LLM to generate these tokens, and extracts a confidence score for each prediction.[4] This confidence-aware fine-tuning provides a practical mechanism for both routing and rejection in LLM cascades.

Beyond single-model confidence estimation, ensemble approaches take a complementary direction by querying multiple models simultaneously. LM-Blender (Jiang et al., 2023) uses an ensemble framework that queries multiple LLMs during inference, selects the best responses, and then merges them into one response. The framework consists of two main modules: Pair Ranker, which uses pairwise comparisons to rank candidate outputs from multiple LLMs, and Gen Fuser, which fuses the top-ranked outputs to generate an improved response. First, the Pair Ranker employs a cross-attention encoder to jointly encode the input and candidate pairs, achieving high correlation with human and ChatGPT-based rankings. Then, the Gen Fuser further enhances output quality by synthesizing the best aspects of top candidates. The approach is evaluated on the MixInstruct benchmark, demonstrating significant improvements over individual LLMs and baseline ensembling methods.

While several cascading approaches restart generation from scratch when a query is escalated to a more capable model, others simply refine the response already generated by the smaller model. In the area of machine translation (MT), cascaded system designs have been extensively studied and deployed in production. Work on automatic post-editing (APE) formulates translation as a two-stage process, in which a base MT system produces an initial hypothesis that is selectively refined by a more capable model when necessary. Usually, quality estimation (QE) models are used to score initial translations, and the routing system decides whether to accept a translation or to escalate to an APE model depending on predefined criteria or thresholds (Specia et al., 2018; Hendy et al., 2023; Moslem et al., 2023a; Zerva et al., 2024). This cascading approach has also been extended to terminology-constrained APE, instructing an LLM to integrate the required terms into the translations (Moslem et al., 2023b). Skipping post-editing for translations that already include pre-approved terms can reduce the overall post-editing load. Such an efficient workflow can save resources, and minimise latency at inference time. Similarly, there are potential advantages of employing an LLM for post-editing rather than for direct translation. Instead of solely relying on the translation quality of the LLM, quality estimation can be performed to select the best MT model in general or for the current source text segment. Ultimately, only segments that do not meet quality criteria are then passed to the LLM for post-editing. Furthermore, QE-based deferral has also been applied directly to routing between small and large translation models, bypassing post-editing entirely. Farinhas et al. (2025) use QE metrics as deferral rules in a cascaded system, invoking the larger model only for a fraction of segments. These cascading approaches demonstrate that quality estimation can effectively guide escalation decisions in hybrid model deployment, reducing large model calls while maintaining quality.

---

[3]The idea of blocking "unsolvable" queries from being routed to the larger model as an efficiency booster has been investigated in detail in recent work such as Firewall Routing (Peng et al., 2025).

[4]To obtain a continuous confidence score after training, the authors compute the probability of the confident token $<CN>$, normalized over the sum of the probabilities of the unconfident $<UN>$ and confident $<CN>$ tokens.

# 8 Multimodal Model Routing

Research on model routing has been focused on text-based LLMs. Many of these approaches can be extended to other modalities such as speech, image, and video. However, representations of these modalities are different from text, which poses new challenges.

Model-Spider (Zhang et al., 2023) is a model selection and ranking method designed to efficiently choose suitable pre-trained models from a large model pool, including both visual models and LLMs, for a downstream task. Unlike the multi-LLM routing mechanisms we covered so far, Model-Spider does not dynamically dispatch queries to different models at inference time. It mainly evaluates candidate models from the available pool and ranks them according to predicted task fitness.

ReLope (Zeng et al., 2026) extends probe-based routing to multimodal LLMs (MLLMs), where visual inputs introduce new challenges for efficient deployment. The authors highlight the limitation that hidden-state probes effective in text-only LLMs degrade substantially when visual inputs are introduced, as visual tokens weaken the separability of correctness signals in hidden states. To address this, they propose two complementary techniques. The Attention Probe aggregates token-level hidden states from the preceding layer using attention scores to recover distributed correctness signals. ReLope (KL-Regularized LoRA Probe) inserts a lightweight LoRA adapter and applies a KL regularizer to learn routing-aware representations that suppress task-irrelevant visual noise. Experiments across five multimodal benchmarks demonstrate improvements over existing probe-based baselines in hybrid MLLM systems.

MMR-Bench (Haoxuan et al., 2026) is a modality-aware evaluation framework. It covers a wide range of vision-language tasks, including OCR, general VQA, and multimodal math reasoning. The benchmark includes strong single-model baselines, oracle upper bounds, and representative routing policies, enabling systematic comparison of routing strategies under fixed candidate sets and cost models.

VL-RouterBench (Huang et al., 2026) is another considerably larger-scale benchmark for VLM routing, covering 14 datasets across three task groups, 30,540 samples, and 17 candidate models (15 open-source and 2 API-based), yielding over half a million sample-model pairs. The benchmark evaluates ten routing methods and baselines, including VLM adaptations of RouterDC (Chen et al., 2024b) and Zooter (Lu et al., 2024) (cf. Section 3), alongside routing architectures specific to the multimodal setting. Its main finding echoes the text-only routing literature: learned routers deliver a significant routability gain over any single model at comparable or lower cost, yet even the best-performing router, an adapted RouterDC, still shows a clear gap to the oracle upper bound, which the authors attribute to the difficulty of jointly modelling fine-grained visual cues and textual structure.

Despite these benchmarking efforts, multimodal routing remains underexplored compared to text-only settings. Key challenges include developing unified representations across modalities, handling queries that require multiple modalities simultaneously, and adapting routing strategies to modality-specific characteristics. Future work should explore how difficulty estimation, uncertainty quantification, and preference alignment can be extended from text-only to multimodal settings.

# 9 Evaluation

In order to optimize routing efficiency, it is imperative to establish the criteria by which routing systems are evaluated. This section reviews commonly used benchmarks and metrics for assessing routing quality, efficiency, and cost, examines how these are applied in practice, and discusses common routing failure modes.

## 9.1 Routing Benchmarks

Several specialized benchmark datasets have been developed to evaluate LLM routing systems systematically.

RouterBench (Hu et al., 2024) has been widely adopted for evaluating routing methods. It provides over 405k precomputed inference outputs from eleven diverse LLMs across seven tasks (MMLU, MT-Bench, MBPP, HellaSwag, WinoGrande, GSM8K, ARC). The dataset includes detailed performance and cost metadata, enabling systematic analysis of routing strategies under realistic constraints.

RouterEval (Huang et al., 2025) is a large-scale open-source benchmark for systematic evaluation of LLM routing methods. The benchmark comprises over 200 million performance records from more than 8,500 LLMs across 12 widely used benchmarks, including ARC, HellaSwag, MMLU, TruthfulQA, GSM8K, and MMLU-PRO. RouterEval frames routing as an m-way classification problem with varying difficulty levels (m = 3, 5, 10, 100, 1000) and supports evaluation across all-strong, all-weak, and mixed model groups. Key findings demonstrate that capable routers can achieve performance surpassing the best single model through model complementarity.

MixInstruct (Jiang et al., 2023) is an instruction-following benchmark designed to evaluate routing and ensemble methods using preference-based supervision. It aggregates 110k examples from multiple instruction datasets and provides oracle pairwise preferences obtained via LLM-based comparisons.

LLMRouterBench (Li et al., 2026) comprises over 400K instances from 21 datasets and 33 models. Moreover, it provides comprehensive metrics for both performance-oriented routing and performance-cost trade-off routing, and integrates 10 routing baselines. The LLMRouterBench framework integrates collector, evaluator, and adaptor components for standardized evaluation of LLM routing methods.

In addition to the aforementioned benchmarks, several further evaluation efforts target specific limitations in existing practice. RouterArena (Lu et al., 2026) introduces an open, leaderboard-style platform for comparing routers across a broad range of knowledge domains and difficulty levels. The DSC benchmark (Kassem et al., 2026) extends evaluation beyond task accuracy to privacy and safety, showing that preference-based routers can make category-driven decisions that inadvertently route jailbreaking attempts to weaker, less-safe models. Yuan et al. (2025) build on RouterBench and EmbedLLM-style query embeddings (cf. Section 2) to argue that existing router benchmarks suffer from limited task diversity, imbalanced model pools, and oversimplified evaluation methodologies, and propose a broader evaluation covering 68 task categories and 85 candidate models.

Beyond routing-specific benchmarks, many routing methods are evaluated on standard LLM benchmarks such as MT-Bench (Zheng et al., 2023) for multi-turn conversations, MMLU (Hendrycks et al., 2021) for knowledge-based reasoning, and MATH-500 (Lightman et al., 2024) and GSM8K (Cobbe et al., 2021) for mathematical problem-solving.

## 9.2 Evaluation Metrics

Evaluating routing systems requires metrics that capture both the quality of routing decisions and their computational efficiency. We organize these metrics into performance, efficiency, and cost metrics.

### 9.2.1 Performance and Quality Metrics

The fundamental goal of routing is to select models that produce high-quality responses. **Routing accuracy** measures the percentage of queries routed to the optimal model. In addition, a confusion matrix identifies false positives and false negatives, which helps calculate the overall performance of the routing system on the current task.

While routing accuracy measures whether the router has selected the "optimal" model for each query, task performance determines whether the selected model has produced a correct or high-quality response. **Task performance** evaluates response quality using domain-specific metrics such as accuracy on multiple-choice questions, exact match for factual queries, pass@k (Kulal et al., 2019; Chen et al., 2021) for code generation, and chrF (Popović, 2017) or COMET (Rei et al., 2020) for translation. Moreover, LLM-as-a-Judge evaluations are increasingly adopted, where a strong LLM ranks the quality of responses from different models. The overall performance of a routing system is the aggregate task performance across all routed queries, effectively treating the router as a meta-system that combines multiple models.

For preference-based routing methods (cf. Section 3), **win rate** measures how often the routed model's response is preferred over a baseline in pairwise comparisons, whether through human judgements or LLM-as-a-Judge evaluations. **Area Under Curve (AUC)** is used to summarise routing performance across different operating points, such as performance across varying cost budgets or deferral thresholds, providing

a single metric for comparing routers independent of specific cost constraints. RouterBench (Hu et al., 2024) introduces **AIQ** (Average Improvement in Quality), which averages the accuracy-cost trade-off curve over a fixed cost range. RouteLLM (Ong et al., 2025) introduces **APGR** (Average Performance Gap Recovered), which integrates the fraction of the weak-to-strong performance gap recovered across cost budgets, and a complementary threshold metric, **CPT($x$%)** (Call-Performance Threshold), the minimum percentage of queries that must be routed to the strong model to recover $x$% of that gap; lower values indicate a more cost-effective router.

### 9.2.2 Efficiency and Cost Metrics

While routing aims to reduce computational costs, it introduces additional system overhead that must be carefully controlled. Obviously, adding larger models to the LLM routing pool contributes to this overhead. Additionally, the overhead resulting from the routing process itself must be negligible.

**Latency** refers to the time between sending a request and receiving a response, and it is commonly measured by the time to first token metric (TTFT) during prefilling and the time per output token metric (TPOT) during decoding (Huyen, 2025).

**Throughput** captures system capacity, and it is measured in tokens per second (TPS) or queries per second (QPS). There is an inherent trade-off between latency and throughput, where serving requests quickly (low latency) often conflicts with serving many requests efficiently (high throughput). **Goodput** measures throughput that meets predefined constraints.[5]

**Cost** represents API charges, compute resources, and token consumption. Typically, cost can be reported relative to strong baselines (e.g., "97% of GPT-4's quality at 24% of the cost") or as a simple monetary value. To jointly evaluate quality and efficiency, quality-cost trade-offs are visualized through **Pareto frontiers** that plot achievable combinations of performance and cost, where effective routers should dominate any single model.

Beyond direct costs, routing systems can reduce environmental impact by favouring smaller models. **Energy consumption** can be reported per token, per query, or as total inference energy, with lower consumption indicating improved efficiency. **Carbon footprint** measures $CO_2$ emissions, estimated from energy consumption and grid carbon intensity. These environmental metrics are increasingly adopted as environmental sustainability becomes central to efficient large-scale deployment.

### 9.3 Evaluation of Routing Systems in Practice

Recent work on multi-LLM routing and cascades has grown rapidly. As part of this effort, researchers have developed specialized routing benchmarks (cf. Section 9.1) or adopted widely used general-purpose LLM benchmarks. Nevertheless, systematic evaluation of routing systems still lacks maturity, and cross-examination of routing approaches remains limited. Tables 2, 3, and 4 demonstrate approaches that rely on RouterBench, MT-Bench, and MixInstruct, respectively. Across the three tables, every method outperforms at least one baseline under its own evaluation protocol, but no metric, model-pool size, or cost basis is shared across rows. On RouterBench, AUC (Cascade Routing), AIQ (GreenServ), accuracy-at-a-cost-point (IRT-Router, MixLLM, PILOT), and area-under-deferral-curve (UniRoute) are different ways of scoring "performance" over pools ranging from an unspecified size up to all 11 RouterBench models. On MT-Bench, the four methods express their gains in different units, namely a quality-drop percentage at a chosen cost-reduction level (BEST-Route), calls-to-target-accuracy as a % of queries (RouteLLM) or converted to USD (Confidence-Driven Router, which adopted RouteLLM's own metric to compare against it directly), and a raw judge score (Zooter). Hence, drawing from the same dataset does not imply a shared evaluation protocol.

---

[5]Efficient inference frameworks such as vLLM (Kwon et al., 2023) offer server versions that can report several standard metrics such as TTFT, TPOT, throughput, and goodput under constraints such as concurrency and maximum token budget.

| Method | Baseline(s) | Performance / Cost | Setup |
|---|---|---|---|
| Cascade Routing | Linear interpolation, Routing, Cascade (baseline) | 87.24% AUC vs. 84.48% for the strongest baseline (11-model, low-noise) | 3/5/11-model subsets tested; cost basis not stated |
| GreenServ | Contextual $\epsilon$-greedy, Thompson sampling | Best accuracy (71.7% avg.) but not best AIQ (0.607 vs. 0.637 for $\epsilon$-greedy) | ∼36k queries, 9 tasks; pool composition not stated |
| IRT-Router | Small/Large LLM only, HybridLLM, RouteLLM, RouterBench's Predictive Router | 80.69% accuracy at $0.55/query vs. 80.01% at $1.15 for RouterBench's Predictive Router | Small=Ministral-8B, Large=GPT-4o; cost=$0.2/M output tokens |
| MixLLM | AutoMix, RouteLLM, Zooter, FORC, OptLLM, MetaLLM, Oracle | 97.25% of GPT-4 quality at 24.18% cost vs. 96.39% at 32.94% for best baseline (OptLLM) | RouterBench + Llama 3.1 8B/70B; 80/20 split |
| PILOT | All-to-one, LinUCB, Epoch-Greedy, HybridLLM | 93% of GPT-4 performance at 25% of its cost, multi-task RouterBench setting (a distinct single-task MMLU-only result reaches 86% at 27% of cost) | 11 LLMs (6 open-source, 5 proprietary) |
| UniRoute | ZeroRouter, K-NN, MLP/Matrix Factorization (oracle) | Two UniRoute variants reported: LearnedMap (0.711, the paper's primary method) and K-Means (0.712, a simpler variant), both vs. 0.707 (K-NN), 0.689 (ZeroRouter); oracle baselines higher (0.720–0.723) | Random 50/50 train/test split of 11 LLMs |

Table 2: Representative methods evaluated on **RouterBench** (Hu et al., 2024), with baselines, results, and experimental setup as reported by each work.

| Method | Baseline(s) | Performance / Cost | Setup |
|---|---|---|---|
| BEST-Route | N-label, N-class, clustering-based routing (all from Srivatsa et al., 2024) | 1.59% quality drop at 60% cost reduction vs. 5.89% for N-label; N-class/clustering never reduce cost | 8-LLM pool; GPT-4o reference; out-of-distribution evaluation |
| Confidence-Driven Router | Random router, TO-Router, RouteLLM | CPT(80%)=55.61 vs. 78.55 for random; $3.74 vs. $4.06 at that operating point | Strong=GPT-4, Weak=Mixtral-8x7B; cost=OpenAI API USD |
| RouteLLM | Random router, BERT, Causal LLM, SW Ranking, Matrix Factorization | Best variant (Matrix Factorization, Arena+Judge-augmented training) needs only 13.40% of queries routed to GPT-4 to recover 50% of the GPT-4-vs.-Mixtral quality gap (CPT(50%)=13.40%) vs. 49.03% for random routing, a 60.4% APGR improvement over random | Strong=gpt-4-1106-preview, Weak=Mixtral 8x7B; cost=% of GPT-4 calls |
| Zooter | BMA (best single model on average), RMR (reward-model ranking; 6 reward-model variants) | 7.11/10 vs. 6.72 for BMA; infers 1 of 6 candidates vs. RMR's 6 | 6x13B LLAMA-based pool; GPT-4 judge |

Table 3: Representative methods evaluated on **MT-Bench** (Zheng et al., 2023), with baselines, results, and experimental setup as reported by each work.

| Method | Baseline(s) | Performance / Cost | Setup |
|---|---|---|---|
| HybridLLM | All-at-large, all-at-small, random | At 20% cost advantage (queries routed to the small model), ≤1% BART-score quality drop for the medium-gap pair (Llama-2-13B→GPT-3.5-turbo); at 40% cost advantage, drop rises to ∼4% for the same pair (up to 10.3% for the large-gap pair, FLAN-T5→Llama-2-13B) — quality drop grows faster than cost savings at higher advantage levels | 3 model pairs tested (small/medium/large performance gap); cost=% of queries routed to the small model (latency/FLOPs/energy explicitly excluded as confounded); GPT-3.5-turbo via OpenAI API |
| LLM-Blender | 11 individual LLMs (best: Open Assistant); 4 metric-oracle upper bounds (BERTScore/ BLEURT/ BARTScore/GPT-Rank); 5 automatic rerankers (Random/MLM-Scoring/SimCLS/ SummaReranker/ PairRanker-alone) | GPT-Rank 3.01 (lower is better) vs. 3.90 for the best single LLM (Open Assistant) | 11-LLM pool (Open Assistant, Vicuna, Alpaca, Baize, MOSS, ChatGLM, Koala, Dolly V2, MPT, StableLM, Flan-T5) |

Table 4: Representative methods evaluated on **MixInstruct** (Jiang et al., 2023), with baselines, results, and experimental setup as reported by each work.

### 9.4 Routing Failure Modes

Understanding how and why routing systems fail is as important as demonstrating where they succeed. We highlight three such modes, namely distribution shift in the query stream, router staleness as the underlying model pool changes, and the asymmetric cost of router errors.

**Distribution shift.** Most routers are trained on a fixed sample of queries and model responses, and their accuracy can degrade when deployed queries diverge from that training distribution. Few surveyed methods are explicitly evaluated under this condition. BEST-Route (Ding et al., 2025) reports an out-of-distribution evaluation, and IRT-Router (Song et al., 2025) addresses unseen queries directly through a semantic similarity warm-up mechanism. For most other methods, robustness to query-distribution shift is untested rather than shown to hold, itself a gap in how routing systems are currently evaluated.

**Router staleness.** A router trained on a fixed model pool becomes stale as that pool changes, whether models are added, removed, or updated by their providers. Several methods are explicitly designed to avoid this, namely GraphRouter (Feng et al., 2024), ICL-Router (Wang et al., 2026), UniRoute (Jitkrittum et al., 2026), and CRE-Router (Moslem et al., 2026); all generalize to new models without retraining the router itself, either through inductive model representations or by automatically re-running a selection procedure (e.g., CRE-Router's Pareto analysis) when the pool changes. By contrast, methods built around a fixed model pool, such as Zooter's reward-distilled classifier, whose limited adaptability to unseen models the original discussion already notes (cf. Section 3), are more exposed to this failure mode. TI-UCB (Xia et al., 2024) addresses a related but distinct form of staleness, non-stationary reward distributions as a single model's performance shifts during iterative fine-tuning, through an explicit change-detection mechanism. Feedback-adaptive methods more broadly carry a related risk beyond technical staleness (cf. Section 12).

**Cost of router errors.** Router mistakes are not symmetric in consequence. Sending a query beyond a weak model's ability risks an incorrect response, while sending a simple query to a strong model only wastes cost and latency. The DSC benchmark's finding that category-driven routers can direct jailbreaking attempts to weaker, less-safe models while reserving the strongest model for already-simple queries (cf. Section 9.1) is a concrete instance of this asymmetry. Most methods surveyed here collapse this distinction into a single blended score, typically an error rate combined linearly with a cost penalty, as in CRE-Router's and UniRoute's cost-adjusted scoring or IRT-Router's performance-cost routing score, which cannot separate a costly-but-safe misrouting from a cheap-but-wrong one. A smaller number of methods address this asymmetry more directly, namely FrugalGPT's cost-aware stop judge (Chen et al., 2024a) and BEST-Route's user-set thresholds (Ding et al., 2025), which let operators tune how conservatively the system escalates, and Firewall Routing (Peng et al., 2025), which specifically targets queries that no model in the pool can answer correctly, where escalation is pure waste rather than a genuine quality-cost trade-off.

## 10 Towards Multidimensional Routing Systems

As we outlined in Section 1.4, the methods reviewed in this survey indicate that practical routing systems are increasingly multidimensional. Rather than selecting a single paradigm, production deployments tend to compose mechanisms across paradigms to satisfy heterogeneous constraints on quality, latency, cost, and safety. They combine pre-generation and post-generation decisions, integrate query-level and response-level signals, and couple offline learning with online adaptation.

One way to design such systems in production is to treat routing as a control pipeline with three coordinated stages: (i) an initial low-cost pre-router based on query and model metadata subject to cost constraints, (ii) a post-generation verifier that estimates response quality or uncertainty of efficient/weaker models, and (iii) an escalation policy that decides whether to accept, refine, reject, or defer to stronger models. In this sense, cascades are a natural extension in routing systems.

To demonstrate how this compositional logic plays out in practice, we can consider representative approaches we have discussed in this survey. For example, both FrugalGPT and Cascade Routing instantiate all three pipeline stages, where a pre-router selects an initial model based on query and cost signals, a quality estimator evaluates the response, and an escalation policy decides whether to accept or defer to a stronger model. Different compositional patterns emerge in bandit-based systems such as MixLLM and PILOT, which couple a pre-generation routing decision with online feedback to continuously refine the policy, effectively collapsing the verifier and escalation stages into an implicit reward signal. Uncertainty-based methods such as CP-Router and Self-REF occupy a middle ground, using post-generation confidence estimates as the escalation trigger without requiring explicit quality estimation models. CRE-Router (cf. Section 7) composes clustering-based routing with a quality-estimation cascade, and is notable for driving both stages from task-correctness labels alone, jointly optimising model selection under an explicit latency budget while recovering accuracy on hard queries without any additional annotation. As illustrated by our design-space matrix (cf. Table 1), these examples suggest that the three-stage pipeline is not a rigid architecture but a flexible template whose stages can be combined, collapsed, or reordered depending on deployment constraints.

This multidimensional perspective also motivates clearer method characterization and comparison. In particular, evaluating routing methods should explicitly report: when decisions are made (pre-generation, post-generation, or multi-stage routing), what signals are used (query, model metadata, response, feedback), and how decisions are computed (heuristic, supervised, bandit, RL policy, or combinations thereof). Methods may activate multiple categories simultaneously, and this compositionality should be reflected in both taxonomy and empirical reporting.

In addition to architectural mapping, our design-space matrix (cf. Table 1) also reveals structural gaps in the current literature that point to concrete directions for future work. First, no current method simultaneously pairs response-level signals with online adaptation, as uncertainty-based approaches and cascades exploit response signals but remain static once deployed, while bandit-based methods adapt online but operate on query-level signals alone. This suggests an opportunity for bandit-style escalation policies that refine thresholds from deployment feedback. Second, reinforcement learning is limited in the cascading paradigm, and combining learned escalation policies with multi-stage architectures remains largely unexplored. Third,

while some systems balance quality, cost, and latency via fixed tradeoff parameters or constraints, few formulate and solve a unified multi-objective optimization problem that treats all metrics as first-class, tunable objectives. Addressing these gaps will be critical for building routing systems that fully exploit the rich design space revealed by the multidimensional framework.

## 11 Conclusion and Future Directions

This survey has covered state-of-the-art approaches to dynamic routing methods for multi-LLM deployment at inference time. We organized approaches into six main paradigms: difficulty-aware routing, human preference alignment, clustering-based methods, reinforcement learning approaches, uncertainty quantification, and cascading systems. Table 5 provides a consolidated summary of all reviewed methods, organised by paradigm. Additionally, we examined emerging work on multimodal routing and discussed evaluation frameworks, including benchmarks and metrics. The key insight is that effective routing systems can outperform even the strongest individual models by strategically leveraging model complementarity and specialization while achieving considerable efficiency gains.

Despite rapid progress, various open challenges remain, with the following potential research directions:

- **Standardized evaluation:** As Section 9 shows, methods sharing the same benchmark rarely share a metric, model pool, or cost basis, which limits direct cross-examination across approaches. We hope this survey encourages the community toward shared evaluation protocols that make routing methods more directly comparable and interpretable.

- **Generalization:** Several routing methods are evaluated on a fixed set of LLMs and struggle to generalize to new models, domains, or data distributions. Developing retraining-free approaches that transfer across diverse architectures, tasks, and deployment scenarios remains an open challenge.

- **Multi-stage cascades:** Most work explores one-stage routing rather than cascading systems (cf. Section 7). As we outlined in Section 1.4 and Section 10, real-world deployments rarely conform to a single paradigm. Future research should reflect real-world applications, where queries and outputs are processed at multiple levels, aiming to strike a balance between quality and efficiency while also ensuring safety.

- **Multimodality:** As models increasingly span vision, audio, and language, routing research should deeply address modality-specific challenges, including fusing multimodal inputs for routing decisions, handling queries requiring multiple modalities, and accounting for varying computational costs of cross-modal architectures.

As the landscape of large foundation models continues to diversify, intelligent routing is becoming increasingly critical for efficient deployment. This survey aims at providing researchers and practitioners with a foundation for understanding current approaches and highlighting opportunities for advancing the state of the art in dynamic multi-LLM systems.

## 12 Ethical Statement

Routing systems that minimize cost by preferring smaller models will, by design, deliver lower-quality outputs for the queries those models handle least well. If the distribution of difficult queries correlates with underrepresented languages, specialized domains, or non-standard writing styles, then cost-optimized routing could quietly introduce quality disparities across user populations without any individual decision reflecting that intent, and speakers of those languages would also bear a higher routing cost, a disparity already observed in other technical contexts such as tokenization efficiency across languages. The feedback-adaptive methods (bandit-based approaches in particular) add a second concern; routing policies that update from deployment feedback can entrench early distributional biases over time. Acknowledging these disparities is a necessary first step toward addressing them.

| Method | Paradigm | Routing Model / Mechanism | Training Data |
|---|---|---|---|
| **BEST-Route** [§2] (Ding et al., 2025) | Difficulty | DeBERTa-v3-small (multi-head) | Question–score pairs |
| **Semantic Router** [§2] (Wang et al., 2025a) | Difficulty | ModernBERT classifier | Query intent labels |
| **RouteLMT** [§2] (Luo et al., 2026) | Difficulty | LoRA probe on small translator representations | Parallel translation pairs (ComMT) |
| **EmbedLLM** [§2] (Zhuang et al., 2024) | Difficulty | Matrix factorization (model embeddings) | Question-answer correctness data |
| **ICL-Router** [§2] (Wang et al., 2026) | Difficulty | LLM router + projector + in-context capability vectors | Query-LLM performance pairs |
| **GraphRouter** [§2] (Feng et al., 2024) | Difficulty | Graph Neural Network (GNN) | Historical performance + cost data |
| **IRT-Router** [§2] (Song et al., 2025) | Difficulty | IRT-based performance predictor (MIRT/NIRT) | Query-LLM performance pairs |
| **RouteLLM** [§3] (Ong et al., 2025) | Preference | Matrix factorization / BERT / Causal LLM | Chatbot Arena + LLM judge |
| **Arch-Router** [§3] (Tran et al., 2025) | Preference | 1.5B parameter LLM | Synthetic preference data |
| **Hybrid LLM** [§3] (Ding et al., 2024) | Preference | LLM-based classifier | Synthetic preference data |
| **P2L** [§3] (Frick et al., 2025) | Preference | LLM (Bradley-Terry coefficients) | Chatbot Arena preferences |
| **Eagle** [§3] (Zhao et al., 2024) | Preference | ELO ranking (training-free) | Historical pairwise comparisons |
| **Zooter** [§3] (Lu et al., 2024) | Preference | mDeBERTa-v3-base classifier | Reward model preference labels |
| **UniRoute** [§4] (Jitkrittum et al., 2026) | Clustering | K-means + unsupervised text embedder | Unlabelled queries + validation set |
| **Avengers-Pro** [§4] (Zhang et al., 2025e) | Clustering | K-means + text embedder + scoring | Labelled query-answer pairs |
| **Router-R1** [§5] (Zhang et al., 2025a) | RL (Policy Opt.) | Qwen2.5-3B / LLaMA-3.2-3B (PPO) | Question–score pairs |
| **R2-Reasoner** [§5] (Shao et al., 2025) | RL (Policy Opt.) | Task Decomposer + Subtask Allocator (GRPO) | SFT + RL training data |
| **MetaLLM** [§5] (Nguyen et al., 2025) | RL (Bandit) | Multi-armed bandit | Online accuracy-cost feedback |
| **MixLLM** [§5] (Wang et al., 2025b) | RL (Bandit) | Contextual bandit + policy gradient | Binary user feedback (online) |
| **PILOT** [§5] (Panda et al., 2025) | RL (Bandit) | LinUCB contextual bandit | Chatbot Arena priors + online feedback |
| **GreenServ** [§5] (Ziller et al., 2026) | RL (Bandit) | LinUCB contextual bandit | Online energy consumption + accuracy feedback |
| **Dueling Feedback** [§5] (Chiang et al., 2025) | RL (Bandit) | CCFT text encoder + FGTS.CDB | Pairwise preference comparisons |
| **TI-UCB** [§5] (Xia et al., 2024) | RL (Bandit) | Time-increasing UCB bandit | Online model performance |
| **CP-Router** [§6] (Su et al., 2025) | Uncertainty | Conformal prediction on logits | Calibration set |
| **LLM-as-a-Judge** [§6] (Zhang et al., 2025c) | Uncertainty | External LLM-as-a-Judge | Human rating preferences |
| **AutoMix** [§7] (Aggarwal et al., 2024) | Uncertainty (Cascade) | POMDP router + few-shot self-verification | Few-shot examples (no fine-tuning) |
| **Self-REF** [§7] (Chuang et al., 2025a) | Uncertainty (Cascade) | Fine-tuned LLM w/ confidence tokens | Correctness-labelled training data |
| **FrugalGPT** [§7] (Chen et al., 2024a) | QE (Cascade) | LLM router + DistilBERT quality estimator | Query-response pairs |
| **Cascade Routing** [§7] (Dekoninck et al., 2025) | QE (Cascade) | Iterative model selection + quality estimator | Query-response pairs (log probabilities) |
| **CRE-Router** [§7] (Moslem et al., 2026) | Clustering (Cascade) | K-means clustering + quality-estimation classifier | Task-correctness labels |
| **LM-Blender** [§7] (Jiang et al., 2023) | Ensemble (Cascade) | Cross-attention Pair Ranker + Gen Fuser | Pairwise preference labels |

Table 5: Summary of LLM routing and cascading methods, reflecting the main paradigms of this survey.

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
