# OpenReview forum: "Dynamic Model Routing and Cascading for Efficient LLM Inference: A Survey"
_TMLR — Under review for TMLR_

### Review · Reviewer_KrUN · 2026-05-06

**Summary Of Contributions:**

The paper tackles a problem that, frankly, has been overdue for a proper treatment. Routing across independently trained LLMs sits in an awkward space: too applied to attract heavy theoretical attention, yet complex enough that practitioners struggle to navigate it without a map. This survey provides that map, and for the most part it does so competently.

The six-paradigm taxonomy is reasonable. One could argue about the boundaries (difficulty-aware routing and uncertainty-based routing overlap more than the structure implies), but for survey purposes the organization holds. What I found more valuable, actually, is the three-dimensional framework introduced in Section 1.4. Characterizing methods along the axes of when, what, and how is simple enough to be memorable and expressive enough to be useful. Table 1 is where this pays off: seeing thirty methods mapped onto the same grid makes gaps in the literature legible in a way that prose alone cannot achieve.

The weaker portions are clustering (two methods; cursory) and multimodality (reads like an afterthought). The survey also inherits a common flaw of the genre. It reports efficiency claims from each paper at face value without adequately flagging that these numbers are not comparable across different baselines, datasets, and task settings. That is worth addressing before publication.

**Additional Comments:**

Several citations carry 2026 dates in a paper currently under review. I assume these are accepted papers or preprints with known publication dates rather than forward citations, but it is disorienting. A brief statement of the survey's coverage window would resolve the confusion.

On balance, this is a well-executed survey of a space that needed one. The framework and the design-space matrix are contributions beyond mere cataloguing. The requested changes above are gaps that can be filled in revision without disrupting the existing structure.

**Audience:**

Yes

**Audience Explanation:**

Without question. Efficient multi-model deployment is pressing enough that researchers across systems, NLP, and ML theory all have stakes in it. The practical framing in Section 10 is useful: decomposing production systems into a pre-router, a post-generation verifier, and an escalation policy is the kind of abstraction that shows up in real engineering discussions, and having it stated crisply in the literature has value.

For researchers specifically, the gaps identified toward the end of Section 10 are worth highlighting. No existing method pairs response-level signals with online adaptation; RL remains largely absent from cascading pipelines; and multi-objective formulations treating quality, cost, and latency as jointly tunable objectives remain underdeveloped. These are not vague gestures toward future work. They follow directly from the design-space matrix, which lends them more credibility than the usual survey hand-waving.

**Broader Impact Concerns:**

There is no Broader Impact statement. For a survey this is probably not a hard requirement, but two issues are worth a brief note. Routing systems that minimize cost by preferring smaller models will, by design, deliver lower-quality outputs for the queries those models handle least well. If the distribution of difficult queries correlates with underrepresented languages, specialized domains, or non-standard writing styles (which there is some reason to expect), then cost-optimized routing could quietly introduce quality disparities across user populations without any individual decision reflecting that intent. The feedback-adaptive methods (bandit-based approaches in particular) add a second concern: routing policies that update from deployment feedback can entrench early distributional biases over time. Neither issue requires extensive treatment, but acknowledging both in a short paragraph would reflect responsible practice and is consistent with TMLR's expectations.

**Claims And Evidence:**

Yes

**Claims Explanation:**

I cross-referenced several method descriptions against their primary sources and found no meaningful misrepresentations. The authors have read the papers carefully. The conceptual framework itself is argued rather than merely asserted, and the mapping in Table 1 is a concrete, checkable instantiation of the design-space claims, which is more than most surveys offer at this level of abstraction.

My main reservation is with quantitative claims inherited from the cited papers. R2-Reasoner's "84.46% API cost savings" is a striking number, but the survey presents it without noting the baseline model, task distribution, or budget constraint under which it was measured. Across the paper there are perhaps a dozen such figures, each drawn from a different experimental setup. Readers who scan the survey for quick comparisons will form misleading impressions. This is less an error of accuracy than of framing. The authors should, at minimum, remind readers consistently that cross-paper numbers are not directly comparable, or better yet, provide a consolidated table that makes the evaluation conditions explicit.

**Requested Changes:**

1. Provide a cross-paper empirical summary. The survey needs, at minimum, a table listing the methods that have been evaluated on shared benchmarks (RouterBench and MT-Bench are the natural candidates), alongside the baseline each paper used and the reported performance-cost trade-off. Without this, the efficiency claims scattered throughout the text are practically uninterpretable. I recognize that experimental conditions vary across papers; the point of the table is to make those variations visible, not to pretend they do not exist.

2. Expand or honestly reframe Section 4. The clustering section covers two methods in under a page, while every other paradigm receives substantially more space. The authors should either locate additional relevant clustering work or explicitly acknowledge in the text that clustering-based routing is less developed than the other paradigms and offer some account of why. Is it harder to evaluate? Is there a theoretical limitation? The current treatment looks incomplete rather than considered.

3. Sharpen the scope boundary discussion. The exclusion of mixture-of-experts is stated once in Section 1.3 and never revisited, yet at least one surveyed method (Router-R1, where the router is itself a large language model performing multi-step reasoning) comes uncomfortably close to the excluded category. A paragraph acknowledging these edge cases and explaining the principle by which they were included or excluded would preempt a predictable reviewer objection and give the scope claim more credibility.

---

> ### Author Response · Authors · 2026-06-28
>
> Thank you for this detailed and constructive review and for the generous closing assessment! Here are responses for the detailed suggestions.
>
> **Cross-paper empirical summary:** This point is now addressed via the three evaluation tables (RouterBench, MT-Bench, MixInstruct) in our central response above, which list each method against the baselines it was actually compared to, with evaluation conditions made explicit.
>
> **Section 4 (clustering):** We will expand this section with additional recent work on clustering-based routing, given the paradigm's importance.
>
> **Scope boundary (MoE exclusion):** We agree that the quick early mention of Mixture of Experts (MoE) in Section 1.3 Scope and Organization might cause confusion. We propose rephrasing the introductory paragraph to clarify that the boundary is based on what the router selects among (independently trained, separately deployable models) rather than how sophisticated the selection mechanism is, and adding a brief discussion of this distinction for the special case of Router-R1 under Section 5.
>
> **Coverage window:** We will add a brief statement clarifying the survey's coverage window. To clarify, cited works carrying 2026 publication dates correspond to their venue's proceedings or to preprints made available in 2026. We believe the high density of recent citations reflects the research community's rapidly growing focus on multi-LLM routing, and we hope our survey serves as a timely and useful reference point for this fast-moving area.
>
> Thank you again for the thorough and constructive engagement with the paper. We agree that the points raised here, together with the cross-paper evaluation tables above, will meaningfully strengthen the revision.

---

> > ### Author Response · Authors · 2026-07-15
> > **Updates**
> >
> > Hello! We have uploaded a new manuscript incorporating the discussed suggestions, as outlined here and in the central comment. If this satisfies your expectations, we will appreciate your positive recommendation. Otherwise, we are happy to continue the discussion. Thanks!

---

### Review · Reviewer_MTwf · 2026-06-07

**Summary Of Contributions:**

This paper surveys dynamic routing and cascading methods for efficient inference across multiple independently trained LLMs. It organizes the literature into six paradigms—difficulty-aware, preference-aligned, clustering-based, RL-based, uncertainty-based, and cascading—and briefly covers multimodal routing and evaluation. Its main conceptual contribution is a three-axis framework characterizing routing systems by when the decision is made, what signals are used, and how it is computed, instantiated in a design-space matrix and used to argue that practical systems are compositional three-stage pipelines (pre-router → verifier → escalation).

Key strengths:
1. The six-paradigm taxonomy is sensible, and the cross-cutting three-axis framework is a genuine organizing contribution that goes beyond a flat list of methods.
2. Coverage is broad and notably current, including very recent (2025–2026) work across multiple subareas.
3. The identification of structural gaps (Section 10) provides actionable directions rather than generic future-work statements.

Key weaknesses:
1. The survey is almost entirely descriptive; it summarizes individual methods but offers limited critical synthesis, quantitative cross-method comparison, or discussion of conflicting empirical findings.

**Audience:**

Yes

**Audience Explanation:**

Efficient multi-LLM deployment is of clear and growing interest to researchers in inference efficiency, ML systems, and practitioners deploying LLMs under cost/latency constraints. The literature is currently fragmented across venues, so a well-organized synthesis fills a real need. The three-axis framework and the explicit list of structural gaps provide a useful reference point and can seed future work.

**Broader Impact Concerns:**

No blocking concerns as the work is efficiency-oriented.

**Claims And Evidence:**

Yes

**Claims Explanation:**

As a survey, the central claims are organizational and synthetic, and these are well supported. The six-paradigm taxonomy is coherent, and the design-space matrix (Table 1) together with the worked examples in §10 convincingly demonstrate the paper's main thesis that practical routing systems are compositional and rarely fit a single paradigm. The method descriptions I cross-checked against the cited works are accurate, and the identification of structural gaps (Section 10) is grounded in the matrix rather than asserted. The framework provides a clear, well-evidenced lens that complements the paradigm-based organization.

**Requested Changes:**

Would strengthen
1. Add a discussion of routing failure modes—distribution shift, router staleness as the model pool changes, cost of router errors—rather than presenting routing mainly through its successes.
2. The three-axis framework is the paper's main contribution but arrives mostly at the end, reading as an add-on. Foreground it earlier and use it to structure the paradigm sections, or add forward references so readers see how each paradigm maps onto the axes.

---

> ### Author Response · Authors · 2026-06-28
>
> Thank you for the positive assessment and the suggestions. We've addressed the shared cross-paper comparability concern raised by all three reviewers in our central response above (mainly the three evaluation tables). Here, we respond to your two specific points.
>
> **Failure modes:** We will add a short subsection discussing routing failure modes, including distribution shift in the query stream, router staleness as the underlying model pool changes, and the asymmetric cost of router errors (misrouting to a weak model vs. unnecessarily escalating to a strong one). This complements the existing discussion of structural gaps in Section 10. We will also note where certain surveyed methods address these issues more explicitly than others, to give readers a sense of which paradigms are better equipped to handle them.
>
> **Foregrounding the three-axis framework:** We will add forward references in Section 1 and under each paradigm section, pointing readers to where that paradigm sits on the when/what/how axes, so the framework is visible throughout rather than only assembled at the end.
>
> Thanks again for the constructive feedback.

---

> > ### Author Response · Authors · 2026-07-15
> > **Updates**
> >
> > Hello! We have uploaded a new manuscript incorporating the discussed suggestions, as outlined here and in the central comment. For example, we added a new section about routing failure modes and referred to the three-axis "design-space framework" throughout the paper. If this satisfies your expectations, we will appreciate your positive recommendation. Otherwise, we are happy to continue the discussion. Thanks!

---

### Review · Reviewer_ufB6 · 2026-06-13

**Summary Of Contributions:**

This is a survey of methods for routing an input task among multiple independently-trained LLMs.
The authors segments methods by whether they route by difficulty, human preferences, unsupervised clustering, or uncertainty, whether the router is trained by RL, whether multi-modal models are routed, and the orthogonal approach of cascading.
The authors discuss design choices about when routing decisions are made and the information used.
Near the end, the authors mention that routing is a multi-objective optimization problem.

Strengths:
* The taxonomy of methods and writing is clear.
* Quantity and relevance of cited work is good.

Weaknesses:
* The bulk of this survey focuses on summarizing the method of each cited routing paper. The summaries fail to explain the pros and cons of each work, and how all the multiple methods in each category relate or compare with one another on common benchmarks.
* The organization of the survey can be improved. You should state the optimization objective of routing up front, and put the evaluation criteria and benchmarks at the very top. This way, while presenting the various routing methods, the metrics and benchmarks can be referenced to compare performance of all the methods that are comparable on shared benchmarks.
* Routing is a multi-objective optimization problem, where someone could have different implicit or explicit weights on accuracy, latency, and cost. This perspective is only lightly mentioned in the survey, whereas it deserves much more emphasis. You should show Pareto plots and state which ones are Pareto-optimal. Even something analogous to the charts in artificialanalysis.ai would improve the paper.

**Audience:**

Yes

**Audience Explanation:**

Assuming that the number of independently-trained models continue to increase, then the routing problem will continue to be relevant going forward, so TMLR readers will be interested.

**Broader Impact Concerns:**

None.

**Claims And Evidence:**

Yes

**Claims Explanation:**

One key claim made in the abstract is:

> We provide a systematic analysis of state-of-the-art multi-LLM routing and cascading approaches.

Note the phrase "state-of-the-art". From the survey, I cannot see which method is the SOTA in all of the method categories. That's because the survey fails to do a cross-examination of the cited methods on the intersection of benchmarks.

Edit 2026/07/17: I updated the answer to 'yes' after checking the author's revised manuscript.

**Requested Changes:**

1. Reorganize the survey to state the optimization problem and benchmarks first.
2. Compare methods to show which is the current SOTA in each category or across categories. You should also re-run some experiments and reproduce methods to set a consensus evaluation if the field is too chaotic and each individual paper evaluates on different tasks.
3. Show Pareto plots for accuracy, latency, cost of methods.

---

> ### Author Response · Authors · 2026-06-28
>
> Thanks for your feedback and suggestions! In addition to the discussion in the central response above, we will include a statement in the Evaluation section noting that some papers had RouterBench/RouterEval available at submission time and opted not to use them, which will serve as evidence for the field's lack of convergence on shared evaluation (rather than as a judgement on those papers' choices). This sits alongside the comparability tables in our central response above, which we believe address the core ask.
>
> We hope this resolves your concern, and would be grateful to know if it does. We are happy to discuss further any other suggestions. Many thanks again for your feedback.

---

> > ### Author Response · Authors · 2026-07-15
> > **Updates**
> >
> > Hello! We have uploaded a new manuscript incorporating the discussed suggestions, as outlined here and in the central comment. For example, we rephrased the second paragraph of the abstract to be clearer about our scope. Also, we added Section 9.3 Evaluation of Routing Systems in Practice to cross-examine approaches across three common benchmarks with three evaluation tables (RouterBench, MT-Bench, MixInstruct). We also highlighted the "standardised evaluation" gap under Section 11 Conclusion and Future Directions. In addition, we added Section 9.4 Routing Failure Modes, including distribution shift, router staleness, and cost of router errors.
> >
> > If this satisfies your expectations, we will appreciate your positive recommendation. Otherwise, we are happy to continue the discussion. Thanks!

---

> > > ### Comment · Reviewer_ufB6 · 2026-07-18
> > > **Thanks for the revision. I have updated the review.**
> > >
> > > Thanks for the revision. I have updated the review.

---

### Comment · Reviewer_KrUN · 2026-06-14
**Discussion: cross-paper comparability, scope, and minor revisions**

Dear authors,

Thank you for your submission. Having read the other two reviews alongside my own, I want to flag that all three of us converge on a single concern, and my recommendation will depend largely on how you propose to address it.
My central reservation is that the survey, while thorough and well-organized in its taxonomy, reports efficiency figures from each cited paper at face value without adequately flagging that these numbers are not comparable across different baselines, datasets, and task settings.
The concrete request is a cross-paper empirical summary: at minimum, a table listing methods evaluated on shared benchmarks (RouterBench and MT-Bench are the natural candidates), the baseline each paper used, and the reported performance–cost trade-off, with evaluation conditions made explicit.
To be clear about the bar: I see two acceptable ways to close the gap between the claims and the evidence. One is the consolidated table above. The other is simply to qualify the claims — in particular, softening the abstract's "state-of-the-art" framing and consistently reminding readers that cross-paper figures are not directly comparable. Either would satisfy my concern; I'd like to understand which path you intend to take before I form a recommendation.

I look forward to your response.

---

### Author Response · Authors · 2026-06-22
**Proposed revisions, mainly a cross-evaluation table**

Thanks to the reviewers for the comments! Please find below our responses to the major suggestions and proposed revisions.

---
- **The abstract's statement, "analysis of state-of-the-art multi-LLM routing and cascading approaches"**

**Observation:** As duly noted by all the reviewers, the survey covers routing papers accepted by top-tier venues from recent years (even 2025 and 2026). When we described these approaches as "state-of-the-art", we were simply referring to this "thorough" coverage. Nevertheless, one year from now, these approaches might not be "state-of-the-art" anymore. In such a case, the survey will mainly act as a reference valid up to its publication date.

**Proposal:**
1. We will remove the adjective "state-of-the-art" from the abstract in the final version to account for long-term validity.
2. We will double-check if there are other distinct approaches that were proposed in the last couple of months, as representative recency was what we mainly aimed to achieve in our survey.

---
- **The concrete request is a cross-paper empirical summary: at minimum, a table listing methods evaluated on shared benchmarks (RouterBench and MT-Bench are the natural candidates), the baseline each paper used, and the reported performance–cost trade-off, with evaluation conditions made explicit.**
---

**Observation:** Model routing is not an accuracy improvement problem; it is a system design choice under constraints that might differ from one setup to another. That is why we focused on system design in our framework. Still, any valid research should have valid evaluation. The clear gap here is cross-approach evaluation, which is implicitly noted by our discussion. Interestingly, although this is the main goal of benchmarks such as RouterBench and RouterEval, several approaches were published before the release of these benchmarks or simply decided not to use them. In addition, even within the same benchmark, most papers do not disclose their serving framework or inference setup, and some have critical differences in objectives (e.g. cost measured in speedup vs monetary spend), which itself prevents assessment of comparability.

**Proposal:**
1. We will add an explicit paragraph to articulate this cross-evaluation gap, hoping our survey can facilitate further interpretability and cross-examination research in this area.
2. We will add the tables suggested by the reviewers under the Evaluation section, one for RouterBench and one for MT-Bench.
3. Since our main contribution (other than the approach coverage) is the design framework, we will make sure it is clearer from the beginning, mainly better connecting Section 1.4 with Section 10.

...

[Due to the character limit of this form, we are continuing the discussion in the following reply.]

---

> ### Author Response · Authors · 2026-06-22
> **[Continued] Evaluation table**
>
> ## RouterBench (Hu et al., 2024)
>
> | Method | Baseline(s) compared | Performance / Cost | Evaluation conditions |
> |---|---|---|---|
> | Cascade Routing (Dekoninck et al., 2025) | Linear interpolation, Routing, Cascade (baseline) | 87.24% AUC vs. 84.48% for the strongest baseline (11-model, low-noise) | 3/5/11-model subsets tested; cost basis not stated |
> | GreenServ (Ziller et al., 2026) | Contextual ε-greedy, Thompson sampling | Best accuracy (71.7% avg.) but not best AIQ (0.607 vs. 0.637 for ε-greedy) | ~36k queries, 9 tasks; pool composition not stated |
> | IRT-Router (Song et al., 2025) | Small/Large LLM only, HybridLLM (Ding et al., 2024), RouteLLM (Ong et al., 2025), RouterBench's Predictive Router | 80.69% accuracy at USD 0.55/query vs. 80.01% at USD 1.15 for RouterBench's Predictive Router | Small=Ministral-8B, Large=GPT-4o; cost = USD 0.2/M output tokens |
> | MixLLM (Wang et al., 2025) | AutoMix (Aggarwal et al., 2024), RouteLLM (Ong et al., 2025), Zooter (Lu et al., 2024), FORC, OptLLM, MetaLLM (Nguyen et al., 2025), Oracle | 97.25% of GPT-4 quality at 24.18% cost vs. 96.39% at 32.94% for best baseline (OptLLM) | RouterBench + Llama 3.1 8B/70B; 80/20 split |
> | PILOT (Panda et al., 2025) | All-to-one, LinUCB (Li et al., 2010), Epoch-Greedy, HybridLLM (Ding et al., 2024) | 93% of GPT-4 performance at 25% of its cost, multi-task RouterBench setting (a distinct single-task MMLU-only result reaches 86% at 27% of cost) | 11 LLMs (6 open-source, 5 proprietary) |
> | UniRoute (Jitkrittum et al., 2026) | ZeroRouter, K-NN, MLP/Matrix Factorization (oracle) | Two UniRoute variants reported: LearnedMap (0.711, the paper's primary method) and K-Means (0.712, a simpler variant) — both vs. 0.707 (K-NN), 0.689 (ZeroRouter); oracle baselines higher (0.720–0.723) | Random 50/50 train/test split of 11 LLMs |
>
> ---
> ## MT-Bench (Zheng et al., 2023)
>
> | Method | Baseline(s) compared | Performance / Cost | Evaluation conditions |
> |---|---|---|---|
> | BEST-Route (Ding et al., 2025) | N-label, N-class, clustering-based routing (all from Srivatsa et al., 2024) | 1.59% quality drop at 60% cost reduction vs. 5.89% for N-label; N-class/clustering never reduce cost | 8-LLM pool; GPT-4o reference; out-of-distribution evaluation |
> | Confidence-Driven Router (Zhang et al., 2025) | Random router, TO-Router, RouteLLM (Ong et al., 2025) | CPT(80%)=55.61 vs. 78.55 for random; USD 3.74 vs. USD 4.06 at that operating point | Strong=GPT-4, Weak=Mixtral-8x7B; cost = OpenAI API USD |
> | RouteLLM (Ong et al., 2025) | Random router, BERT, Causal LLM, SW Ranking, Matrix Factorization | Best variant (Matrix Factorization, Arena+Judge-augmented training) needs only 13.40% of queries routed to GPT-4 to recover 50% of the GPT-4-vs-Mixtral quality gap (CPT(50%)=13.40%) vs. 49.03% for random routing, a 60.4% APGR improvement over random | Strong=gpt-4-1106-preview, Weak=Mixtral 8x7B; cost = % of GPT-4 calls |
> | Zooter (Lu et al., 2024) | BMA (best single model on average), RMR (reward-model ranking; 6 reward-model variants) | 7.11/10 vs. 6.72 for BMA; infers 1 of 6 candidates vs. RMR's 6 | 6×13B LLAMA-based pool; GPT-4 judge |
>
> ---
>
> ### Metrics used (addition to the Metrics subsection)
>
> - **AUC** (Cascade Routing): area under the accuracy-cost trade-off curve, computed across multiple operating points; higher indicates a better trade-off.
> - **AIQ** (GreenServ): RouterBench's own cost-performance trade-off metric, averaged across willingness-to-pay operating points.
> - **APGR** (RouteLLM): Average Performance Gap Recovered, like AUC, an integral over cost budgets, but of a normalized weak-to-strong gap rather than raw quality.
> - **CPT(x%)** (Confidence-Driven Router): minimum fraction of queries that must be routed to the strong model to recover x% of the accuracy gap over the weak model alone; lower is more cost-effective.
> - **Area-under-deferral-curve** (UniRoute): the area under the curve relating the fraction of queries deferred to a larger model against resulting accuracy; higher indicates a better accuracy-deferral trade-off.
>
> ---
>
> ### Takeaways
>
> Across both tables, every method outperforms at least one baseline under its own paper's protocol, but no metric, model-pool size, or cost basis is shared across rows. On RouterBench: AUC (Cascade Routing), AIQ (GreenServ), accuracy-at-a-cost-point (IRT-Router, MixLLM, PILOT), and area-under-deferral-curve (UniRoute) are different ways of scoring "performance" over pools ranging from an unspecified size up to all 11 RouterBench models. On MT-Bench: the four methods express their gains in different units: a quality-drop percentage at a chosen cost-reduction level (BEST-Route), calls-to-target-accuracy as a % of queries (RouteLLM) or converted to USD (Confidence-Driven Router, which adopted RouteLLM's own metric to compare against it directly), and a raw judge score (Zooter). Hence, drawing from the same dataset does not imply a shared evaluation protocol, which reveals a gap and opportunity for future work.

---

> ### Author Response · Authors · 2026-06-22
> **[Continued] Third table for MixInstruct benchmark**
>
> ## MixInstruct (Jiang et al., 2023)
>
> | Method | Baseline(s) compared | Performance / cost (headline) | Evaluation conditions |
> |---|---|---|---|
> | HybridLLM (Ding et al., 2024) | All-at-large, all-at-small, random | At 20% cost advantage (queries routed to the small model), ≤1% BART-score quality drop for the medium-gap pair (Llama-2-13B→GPT-3.5-turbo); at 40% cost advantage, drop rises to ~4% for the same pair (up to 10.3% for the large-gap pair, FLAN-T5→Llama-2-13B) — quality drop grows faster than cost savings at higher advantage levels | 3 model pairs tested (small/medium/large performance gap); cost = % of queries routed to the small model (latency/FLOPs/energy explicitly excluded as confounded); GPT-3.5-turbo via OpenAI API |
> | LLM-Blender (Jiang et al., 2023) | 11 individual LLMs (best: Open Assistant), 4 oracles (BERTScore/BLEURT/BARTScore/GPT-Rank), Random/MLM-Scoring/SimCLS/SummaReranker/PairRanker-alone rankers | GPT-Rank 3.01 (lower is better) vs. 3.90 for the best single LLM (Open Assistant) | 11-LLM pool (Open Assistant, Vicuna, Alpaca, Baize, MOSS, ChatGLM, Koala, Dolly V2, MPT, StableLM, Flan-T5) |

---

> ### Comment · Reviewer_ufB6 · 2026-06-28
> **Critical evaluation would increase the utility of the survey**
>
> Of course a survey is not expected to be relevant $n$ years from now, but that does not mean one can shirk away from the responsibility of stating the SOTA at the present time. Stating the SOTA and critically comparing methods on the same benchmark would make a survey more relevant for researchers in the field to build intuition on the relative effectiveness of different classes of methods.
>
> Regarding:
> >  several approaches were published before the release of these benchmarks or simply decided not to use them. In addition, even within the same benchmark, most papers do not disclose their serving framework or inference setup, and some have critical differences in objectives (e.g. cost measured in speedup vs monetary spend), which itself prevents assessment of comparability.
>
> If this is the case, then you should directly point it out. Papers that seem to deliberately avoid some benchmarks should be pointed out as problematic. Differences in experimental protocol among existing work needs to be disclosed and emphasized, instead of glossed over or avoided in a survey. That is the whole point of a survey, to bring out cross-paper issues that is not apparent from looking at papers in isolation.

---

> > ### Author Response · Authors · 2026-06-28
> > **Benchmarks and Critical Assessment**
> >
> > Thanks to the reviewer for the comment! We were talking specificly about routing benchmarks such as RouterBench and RouterEval. Some papers used other benchmarks, which is still valid to compare with their baselines. Upon the recommendation of other reviewers, we have created and proposed adding the evaluation tables above, which cross-examine approaches (first column) with baselines (second column) under shared conditions.
> >
> > As for critical assessment, we believe the main limitation in the field is studying individual routing aspects rather than multi-stage end-to-end systems common in real-world production. This is exactly what we highlighted throughout our survey and more elaborately in Section 10 and Table 1.
> >
> > We thank the reviewers again for their constructive feedback. If there are any other suggestions, we will be happy to listen to them. Thanks!

---

> ### Author Response · Authors · 2026-06-28
> **Update**
>
> Update: Following the discussion above, we have also replied individually to the reviewers on points specific to their reviews.
>
> We are very glad to keep iterating on any of this. Please let us know if our responses address your concerns or if there are other points you would like us to look into. Many thanks again for all the feedback!

---

> ### Comment · Reviewer_KrUN · 2026-06-30
> **Tables resolve central concern; two items before I finalize**
>
> Thank you for the thorough and responsive engagement with my review, and for addressing the concern all three of us converged on directly rather than deflecting it.
> On my central reservation: the three evaluation tables (RouterBench, MT-Bench, MixInstruct) resolve it. They do exactly what I asked, listing each method against the baselines it was actually compared to with evaluation conditions made explicit, and the "Takeaways" synthesis is more valuable than I expected, since it turns the non-comparability from an unstated caveat into an articulated finding about the field's lack of shared protocol. Combined with dropping "state-of-the-art" from the abstract and the commitment to consistently remind readers that cross-paper figures are not directly comparable, both of the paths I outlined have been taken, which more than satisfies the bar I set.
> My other requested changes are also addressed satisfactorily in principle: the Section 4 clustering expansion, and the MoE scope reframing around what the router selects among rather than the sophistication of the mechanism, with a Router-R1 discussion in Section 5. The latter is the right principle and should preempt the predictable objection.
> Two things I'd ask before I finalize a positive recommendation:
> First, please upload a revised manuscript incorporating these changes. At present the tables, the abstract edit, and the section revisions are committed in this discussion thread rather than visible in the paper, and I'd like to confirm they land in the manuscript as described, in particular that the three tables and the non-comparability framing are integrated into the Evaluation section rather than appearing only here.
> Second, on broader impact: my review raised two specific concerns (cost-optimized routing quietly introducing quality disparities across user populations, and feedback-adaptive bandit methods entrenching early distributional biases). These were not addressed in your reply to me. I'm not asking for extensive treatment, but a short paragraph acknowledging both would reflect responsible practice and is consistent with TMLR's expectations. Please confirm whether you intend to add it.
> Subject to the revised manuscript reflecting the commitments above, I am inclined to recommend acceptance. The taxonomy, the three-axis framework, and the design-space matrix are genuine contributions beyond cataloguing, and the requested changes are gap-filling rather than structural. I look forward to the revision.

---

> > ### Author Response · Authors · 2026-06-30
> > **Manuscript (including Ethical Statement)**
> >
> > Many thanks for your response and potential recommendation!
> > - Yes, we will shortly upload a new manuscript to incorporate the proposed changes.
> > - Regarding the Broader Impact Concerns, we agree on what you suggested about quality disparities and distributional biases. Also, if hard queries represent certain languages, this means that some language speakers will have to pay more for these queries, which we already see in other technical cases like tokenizers. We will add an Ethical Statement section to discuss these issues. I do not know the solution to such disparities, but the first step to solve any issue is to acknowledge it, hopefully to draw the required attention that leads to a solution.
> >
> > Thanks again for your time and constructive feedback! We highly appreciate it.

---

> ### Author Response · Authors · 2026-07-13
> **Revised Manuscript Uploaded**
>
> Hello! As per the discussion with the reviewers, we have now uploaded a revised manuscript incorporating the following major changes:
> - Two new subsections have been added, among other changes, under section 9 Evaluation, namely:
>   - Added subsection 9.3 Evaluation of Routing Systems in Practice, including three benchmark tables and a takeaway introduction, as per the aforementioned discussion. (in response to all reviewers KrUN, MTwf, and ufB6)
>   - Added subsection 9.4 Routing Failure Modes, including distribution shift, router staleness, and cost of router errors. (in response to reviewer MTwf)
>   - Removed outperformance claims in the survey text itself inherited from papers, beyond section 9.3 tables. (in response to reviewer KrUN)
>   - Updated subsection 9.2.1 Performance and Quality Metrics with more metrics from the tables.
>   - Updated subsection 9.1 Routing Benchmarks with other recently released routing benchmarks.
>
> - Connected Section 1.4 and Section 10. Also, added forward references in Section 1 connecting to Section 10 and under each paradigm section, pointing readers to where that paradigm sits on the when/what/how axes, so the framework is visible throughout rather than only assembled at the end. (in response to reviewer MTwf)
>
> - Scope boundary: Rephrased the first paragraph of Section 1.3 Scope and Organization to avoid the confusion about MoE. Also, added a sentence to 5.1 Policy Optimization Methods, clarifying that “Although the router in these methods is itself an LLM performing multi-step reasoning, all methods in this section dispatch queries to independently trained, separately deployed models, so the routing happens between model boundaries rather than within a shared architecture.”  (in response to reviewer KrUN)
>
> - Extended Section 4 Clustering-based Routing to include two new approaches, Avengers and CRE-Router. As the latter approach, CRE-Router, is also a cascaded approach, it is further discussed in Section 7 Cascades.  (in response to reviewer KrUN)
>
> - Added Section 12 Ethical Statement, highlighting some inequities and biases that arise with efficient approaches in general and routing systems in particular. (in response to reviewer KrUN)
>
> The strengths outlined by the reviewers remain intact, including:
> - The paper tackles a problem that has been overdue for a proper treatment, i.e. routing across independently trained LLMs, which makes it also relevant to TMLR's audience. (reviewer KrUN)
> - The sensible six-paradigm taxonomy and the cross-cutting three-axis framework as a genuine organizing contribution that goes beyond a flat list of methods. (reviewer MTwf)
> - Quantity and relevance of cited work; broad coverage that is notably current. (reviewers ufB6 and MTwf)
>
>
> With this, we would like to thank the reviewers for their constructive feedback and astute suggestions that have helped the paper become even clearer and more beneficial to the readers. We highly appreciate this, and we are happy to continue the discussion.

---

### Comment · Reviewer_KrUN · 2026-07-22
**All conditions met; Accept + Survey Certification confirmed**

I've checked the revised manuscript and confirm the committed changes are incorporated as described: the three evaluation tables with the non-comparability framing (Section 9.3), the abstract edit, the Section 4 clustering expansion, the MoE scope clarification (Sections 1.3 and 5.1), and the Ethical Statement (Section 12) covering the broader-impact concerns I raised. The additions beyond my requests — Section 9.4 on routing failure modes and the removal of inherited outperformance claims from the survey text — further strengthen the paper. My Accept + Survey Certification recommendation stands unconditionally. Thank you for the thorough and responsive revision.